# Analyses of child cardiometabolic phenotype following assisted reproductive technologies using a pragmatic trial emulation approach

Jonathan Yinhao Huang [1✉], Shirong Cai[1,2], Zhongwei Huang [2,3], Mya Thway Tint [1,2], Wen Lun Yuan[1,4], Izzuddin M. Aris [5], Keith M. Godfrey [6], Neerja Karnani [1], Yung Seng Lee[1,7], Jerry Kok Yen Chan[8,9], Yap Seng Chong[1,2], Johan Gunnar Eriksson[1,2,10,11] & Shiao-Yng Chan [1,2]

Assisted reproductive technologies (ART) are increasingly used, however little is known about the long-term health of ART-conceived offspring. Weak selection of comparison groups and poorly characterized mechanisms impede current understanding. In a prospective cohort (Growing Up in Singapore Towards healthy Outcomes; GUSTO; Clinical Trials ID: NCT01174875) including 83 ART-conceived and 1095 spontaneously-conceived singletons, we estimate effects of ART on anthropometry, blood pressure, serum metabolic biomarkers, and cord tissue DNA methylation by emulating a pragmatic trial supported by machine learning-based estimators. We find ART-conceived children to be shorter (−0.5 SD [95% CI: −0.7, −0.2]), lighter (−0.6 SD [−0.9, −0.3]) and have lower skinfold thicknesses (e.g. −14% [−24%, −3%] suprailiac), and blood pressure (−3 mmHg [−6, −0.5] systolic) at 6-6.5 years, with no strong differences in metabolic biomarkers. Differences are not explained by parental anthropometry or comorbidities, polygenic risk score, breastfeeding, or illnesses. Our simulations demonstrate ART is strongly associated with lower *NECAB3* DNA methylation, with negative control analyses suggesting these estimates are unbiased. However, methylation changes do not appear to mediate observed differences in child phenotype.

---

[1] Singapore Institute for Clinical Sciences (SICS), Agency for Science, Technology, and Research (A*STAR), Singapore, Singapore. [2] Department of Obstetrics and Gynaecology, Yong Loo Lin School of Medicine, National University of Singapore, Singapore, Singapore. [3] Institute of Molecular and Cell Biology (IMCB), Agency for Science, Technology, and Research (A*STAR), Singapore, Singapore. [4] Université de Paris, CRESS, Inserm, Paris, France. [5] Division of Chronic Disease Research Across the Lifecourse, Department of Population Medicine, Harvard Medical School and Harvard Pilgrim Health Care Institute, Boston, MA, USA. [6] MRC Lifecourse Epidemiology Centre and NIHR Southampton Biomedical Research Centre, University of Southampton and University Hospital Southampton, Southampton, UK. [7] Department of Paediatrics, Yong Loo Lin School of Medicine, National University of Singapore, Singapore, Singapore. [8] Department of Reproductive Medicine, KK Women's and Children's Hospital, Singapore, Singapore. [9] Academic Clinical Program in Obstetrics and Gynaecology, Duke-NUS Medical School, Singapore, Singapore. [10] University of Helsinki, Department of General Practise and Primary Health Care, Helsinki University Hospital, Helsinki, Finland. [11] Folkhälsan Research Center, Helsinki, Finland. ✉email: jonathan_huang@sics.a-star.edu.sg

Assisted reproductive technologies (ART) such as intracytoplasmic sperm injection (ICSI) and in vitro fertilization (IVF) are increasingly common worldwide, resulting in hundreds of thousands of live births each year[1]. In the United States, 182,111 procedures were performed in 2015, accounting for 1.7% of all live births[2]. In Singapore, ART (predominantly ICSI-IVF) cycles exceed 5000 annually and are increasing[3]. Though ART has been associated with a number of perinatal and infant morbidities including preterm birth and low birth weight[4], longer-term effects on offspring remain unclear[5,6]. A recent systematic review of offspring growth found IVF-conceived children appeared slightly lighter than spontaneously conceived (SC) at younger ages (birth to 4 years), but not at 5+ years, with little evidence of differences in height[7]. However, the review highlighted major limitations of prevailing studies including lack of accounting for parental anthropometry or fertility status, paternal characteristics, and differential child measurement protocols. Importantly, comparison groups varied greatly, from convenience samples of volunteers to children born following other fertility treatments (e.g., ovarian stimulation). Statistically, previous studies also suffered from biases due to inappropriate overadjustment for birth weight and other mediators, failure to account for differential loss-to-follow-up, and limited numbers of evaluation time points. Evidence for cardiometabolic effects is also sparse: small studies in adolescents have reported some evidence of increased peripheral adiposity[8], vascular dysfunction[9], and elevated blood pressure and fasting glucose[10]. However, others have found no such relationships[11]. A study comparing 54 ART-conceived adolescents to 43 age- and sex-matched SC found the former to have higher 24-h ambulatory blood pressures[12].

Moreover, while ART procedures expose blastocysts, trophoblasts, and developing fetuses[13,14] to environmental exposures linked to epigenetic reprogramming[15], particularly at imprinted genes[11,16,17], putative mechanisms in humans remain elusive[4,11–13]. Notably, while persistent fetal epigenetic modification may link known risk factors for infertility such as maternal overweight[18] to placental[19,20] and early metabolic[21] pathologies, and to offspring growth and metabolic capacity, these links have not been synthesized in the ART context. For example, maternal obesity may lead to reduced hypoxia-inducible factor 1 (HIF1A) expression in both maternal (follicular maturation[22]) and fetal (poor trophoblast invasion, fetal growth[17], and metabolic dysregulation[19]) tissues, but their role in the growth of ART children is unknown. Recent EWAS show suggestive associations with ART[23], but such associations may not persist longitudinally[24] and are likely confounded by infertility. Overall, the growth and potential cardiometabolic effects of ART among children and likely pathophysiologic mechanisms are not well-characterized.

The Growing Up in Singapore Towards healthy Outcomes (GUSTO) prospective mother–offspring cohort provides an opportunity to more deeply investigate and address these gaps. In GUSTO, mothers who conceived spontaneously and through ART, predominantly ICSI-IVF, were similarly recruited in the first trimester from the two major national obstetric clinics ($N = 83$ ART-conceived and 1095 spontaneously conceived) and assessments were conducted without regard to the mode of conception. Offspring were followed up for differences in anthropometry (height, weight, BMI, and skinfold thickness), body composition (lean and fat mass) measured by quantitative magnetic resonance imaging (QMR)[25], and blood pressure from birth through 6.5 years as well as serum cardiometabolic measures at 6 years (e.g., fasting glucose and insulin, cholesterol and triglycerides, liver enzymes). Importantly, it is possible to deploy newer design and analytic methods and best practices for nonrandomized studies of intervention[26] to addresses previous gaps. Namely, target pragmatic trial amongst couples who may be seeking fertility consultations may be emulated by constructing putatively subfertile, but spontaneously conceiving, comparison cohorts along with implementing doubly robust, cross-validated, ensemble machine-learning estimators for treatment effects. To investigate mechanisms, simulation modeling within these emulate trials may be used to explore the hypothesis that ART-associated perinatal and postnatal factors such as maternal hyperglycemia and hypertension; differential offspring DNA methylation; or breastfeeding duration[27] causally mediate any observed ART effects. Furthermore, negative control approaches simulating mediation by maternal epigenome may be used to detect the presence of residual confounding bias due to unmeasured maternal genetic and environmental factors that may influence both fetal epigenome and child growth.

In this study, we estimate the effects of ART-assisted conception on child anthropometry and cardiometabolic outcomes using several complementary designs and statistical modeling strategies with different underlying assumptions (described in "Methods"), taking into account differential treatment probabilities, residual confounding, and loss-to-follow-up. We find evidence for reduced height, weight, skinfold thickness, and blood pressure at 6–6.5 years, but weak evidence for changes in metabolic biomarkers. Focusing specifically on candidate CpGs ($N = 281$) previously identified by related EWAS (Supplemental Data), we find evidence for unconfounded effects on *NECAB3* CpGs, however little evidence for a mediating role. These findings appear to be reassuring regarding child health sequelae of ART; however, they require replication in larger consortia efforts.

## Results

**Parental sociodemographic characteristics and medical history**. Parents of ART-conceived and SC children differed in anticipated ways (Table 1). Notably, parents of ART-conceived children were older, more likely to be nulliparous, had higher income and education, and reported less frequent exposure to tobacco smoke in the home during pregnancy. ART fathers were more likely to report a history of diabetes or high blood pressure prior to pregnancy but were of similar height and weight. Maternal glucose concentrations 2-h after a 75-g oral glucose tolerance test (OGTT) at 26–28 weeks gestation were higher among mothers who conceived via ART, in line with a previous report from this cohort[28]. Mothers who conceived via ART also had higher weight and a higher self-reported prevalence of any current or past elevated blood pressure (Chronic HTN, PIH, PE, or eclampsia; Table 1).

To emulate a target pragmatic trial[21], we formed comparison ("control") arms of spontaneously conceived children by sampling from the overall study population based on parental medical history (e.g., polycystic ovarian syndrome) and medications suggestive of infertility risk (primary arm $N = 93$). Without explicitly matching on demographic characteristics or other comorbidities, this subcohort was effectively more similar to the ART-conceived group with respect to parental sociodemographic characteristics, maternal glucose concentrations, and blood pressure than the general study population, including paternal anthropometry and medical history (Supplemental Table 1).

**Offspring anthropometry, body composition, and cardiometabolic risk factors**. Children conceived via ART were similar in terms of gestational age at birth, mode of delivery, birth weight, and birth length, with a difference in height (2 cm shorter) apparent by 5 years of age regardless of comparison arm (Supplemental Table 2). Lower weight and BMI and shorter height

**Table 1 Overall study population parental characteristics, by conception status[a].**

| | Spontaneous conception, all (n = 1095) | Spontaneous, possibly subfertile[b] (n = 93) | ART (n = 83) |
|---|---|---|---|
| Mother's age at delivery (years) | | | |
| Mean (SD) | 30.9 (±5.13) | 33.5 (±4.67) | 34.5 (±2.97) |
| Nulliparous | | | |
| Yes | 468 (42.7%) | 38 (40.9%) | 67 (80.7%) |
| No | 627 (57.3%) | 55 (59.1%) | 16 (19.3%) |
| Mother's ethnicity | | | |
| Chinese | 598 (54.6%) | 67 (72.0%) | 64 (77.1%) |
| Malay | 291 (26.6%) | 12 (12.9%) | 8 (9.6%) |
| Indian | 205 (18.7%) | 14 (15.1%) | 11 (13.3%) |
| Other | 1 (0.1%) | 0 (0%) | 0 (0%) |
| Mother's highest educational qualification | | | |
| No education | 2 (0.2%) | 0 (0%) | 0 (0%) |
| Primary | 58 (5.3%) | 5 (5.4%) | 0 (0%) |
| Secondary | 282 (25.8%) | 14 (15.1%) | 13 (15.7%) |
| ITE/NTC | 112 (10.2%) | 1 (1.1%) | 4 (4.8%) |
| GCE A-Level | 272 (24.8%) | 21 (22.6%) | 24 (28.9%) |
| University | 355 (32.4%) | 52 (55.9%) | 41 (49.4%) |
| Missing | 14 (1.3%) | 0 (0%) | 1 (1.2%) |
| Monthly household income (SGD) | | | |
| 0–999 | 23 (2.1%) | 1 (1.1%) | 0 (0%) |
| 1000–1999 | 140 (12.8%) | 8 (8.6%) | 4 (4.8%) |
| 2000–3999 | 324 (29.6%) | 17 (18.3%) | 12 (14.5%) |
| 4000–5999 | 255 (23.3%) | 27 (29.0%) | 18 (21.7%) |
| 6000+ | 282 (25.8%) | 38 (40.9%) | 41 (49.4%) |
| Missing | 71 (6.5%) | 2 (2.2%) | 8 (9.6%) |
| Mother's height (cm) | | | |
| Mean (SD) | 158 (±5.64) | 159 (±5.64) | 159 (±5.56) |
| Missing | 29 (2.6%) | 1 (1.1%) | 2 (2.4%) |
| Mother's pre-pregnancy BMI (kg/m$^2$) | | | |
| Mean (SD) | 22.7 (±4.42) | 22.4 (±3.73) | 23.4 (±4.78) |
| Missing | 102 (9.3%) | 5 (5.4%) | 8 (9.6%) |
| Maternal fasting glucose (mmol/L) | | | |
| Mean (SD) | 4.35 (±0.468) | 4.34 (±0.408) | 4.48 (±0.642) |
| Missing | 53 (4.8%) | 3 (3.2%) | 5 (6.0%) |
| Maternal 2-h post-OGTT glucose (mmol/L) | | | |
| Mean (SD) | 6.50 (±1.46) | 6.85 (±1.45) | 7.21 (±1.75) |
| Missing | 53 (4.8%) | 3 (3.2%) | 5 (6.0%) |
| Maternal high blood pressure history | | | |
| No | 854 (78.0%) | 66 (71.0%) | 60 (72.3%) |
| Yes | 62 (5.7%) | 7 (7.5%) | 11 (13.3%) |
| Missing | 179 (16.3%) | 20 (21.5%) | 12 (14.5%) |
| Any smoking in home (during pregnancy) | | | |
| No | 676 (61.7%) | 72 (77.4%) | 64 (77.1%) |
| Yes | 368 (33.6%) | 17 (18.3%) | 15 (18.1%) |
| Missing | 51 (4.7%) | 4 (4.3%) | 4 (4.8%) |
| Father's age at delivery (years) | | | |
| Mean (SD) | 34.3 (±5.91) | 35.7 (±5.07) | 39.1 (±4.64) |
| Missing | 386 (35.3%) | 25 (26.9%) | 26 (31.3%) |
| Father's height (cm) | | | |
| Mean (SD) | 171 (±6.17) | 172 (±5.48) | 172 (±5.96) |
| Missing | 424 (38.7%) | 32 (34.4%) | 31 (37.3%) |
| Father's weight (kg) | | | |
| Mean (SD) | 75.4 (±14.3) | 77.6 (±13.3) | 76.0 (±15.5) |
| Missing | 427 (39.0%) | 32 (34.4%) | 31 (37.3%) |
| Paternal diabetes | | | |
| No | 706 (64.5%) | 69 (74.2%) | 53 (63.9%) |
| Yes | 18 (1.6%) | 1 (1.1%) | 4 (4.8%) |
| Missing | 371 (33.9%) | 23 (24.7%) | 26 (31.3%) |
| Paternal high blood pressure history | | | |
| No | 651 (59.5%) | 60 (64.5%) | 45 (54.2%) |
| Yes | 72 (6.6%) | 10 (10.8%) | 12 (14.5%) |
| Missing | 372 (34.0%) | 23 (24.7%) | 26 (31.3%) |

*SGD* Singapore dollars, *cm* centimeters, *m* meters, *kg* kilograms.
[a]Cells presented as N (%) unless otherwise specified as mean (SD).
[b]A subset of all spontaneous conceptions (first column) selected on the basis of parental medical history and medications indicative of difficulty conceiving.

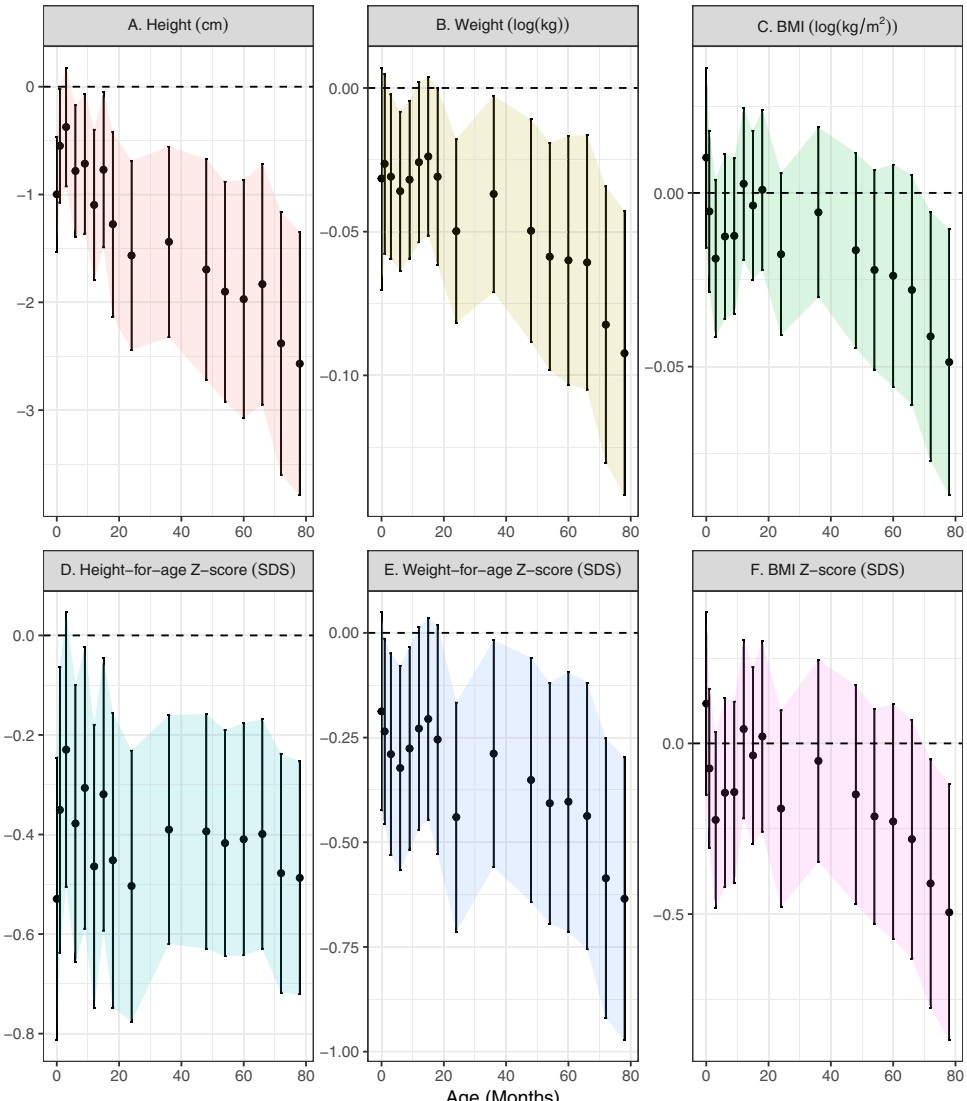

**Fig. 1 Associations between ART status and anthropometry, adjusted for pre-pregnancy characteristics.** Point estimates and Wald-type 95% confidence intervals show the adjusted mean difference in anthropometric measures between ART-conceived and spontaneously conceived (SC) children at each study visit, with SC children as the referent. Differences were estimated by multivariable linear regressions adjusted for maternal age, education, ethnicity, household income, height, pre-pregnancy BMI, parity, and any tobacco exposure in the home; paternal age, height, weight, diabetes history, and high blood pressure history; child sex, and polygenic risk score for adult adiposity with no multiple testing adjustments. Multiple imputations by chained equations were used to estimate associations while simultaneously accounting for missing covariate values. Sample sizes at each visit (SC/ART): 1091/83; 967/69; 953/71; 916/68; 872/70; 892/68; 874/67; 807/55; 832/60; 864/64; 801/57; 833/64; 812/61; 800/63; 771/59; 762/59. Shaded regions connect confidence intervals to aid in visualizing trends and provide no additional information regarding estimates between visits. cm centimeters, kg kilograms, SDS standard deviation scores.

among ART children tended to be more pronounced after 36 months, but patterns varied by anthropometric measure (Fig. 1).

Ultrasound-estimated fetal weights did not differ between ART-conceived and SC children at earlier gestational ages. ART-conceived children had a slightly heavier estimated fetal weight (90 g) at ~32 weeks, however, adjusted estimates were imprecise (Supplemental Fig. 1). In adjusted models, postnatal height, weight, and to a lesser extent BMI were consistently lower among ART-conceived children after 18–24 months. This was true using either absolute or age- and sex-standardized measures (Fig. 1). At 6.5 years, children conceived by ART were on average 0.5 SD shorter [95% confidence interval: −0.7, −0.2] and 0.6 SD lighter [−1.0, −0.3] (Fig. 1). Children conceived by ART had similar or

thinner skinfolds (e.g., 18% thinner subscapular skinfolds [−27%, −8%] at 6.5 years), and lower systolic and diastolic blood pressures (−3.2 mmHg [−5.8, −0.5] and −2.3 mmHg [−4.3, −0.5], respectively at 6 years; Fig. 2).

Models adjusted for conditions of pregnancy and delivery including maternal fasting and post 2-h OGTT glucose concentrations at 26 weeks, maternal elevated blood pressure in pregnancy, and gestational age at delivery did not change estimates (Table 2 presents representative results at 6 years, since blood pressures were not measured at 6.5 years).

In a subsample with body composition measured by QMR at 5 and 6 years of age (N = 247 and 379, respectively), ART-conceived children had lower fat mass than SC children but similar lean mass (Supplemental Table 3 and Supplemental

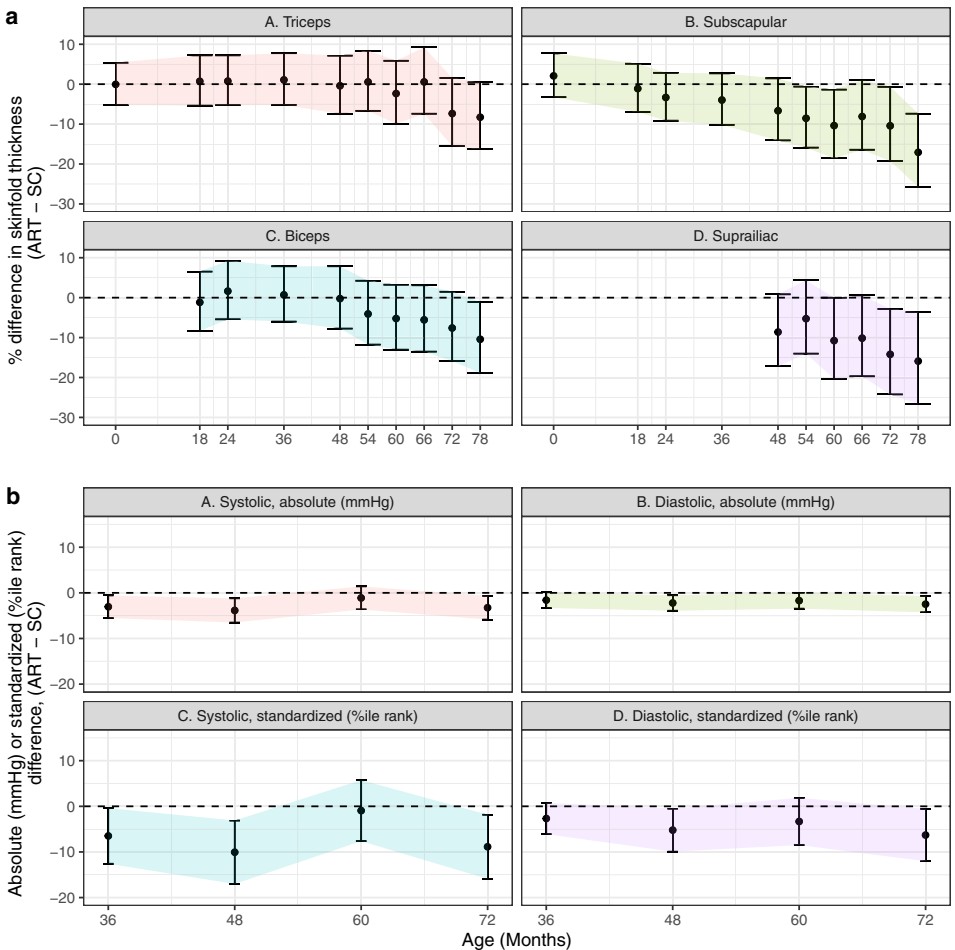

**Fig. 2 Associations between ART status and skinfold thickness and blood pressure, adjusted for pre-pregnancy characteristics.** Point estimates and Wald-type 95% confidence intervals show the adjusted mean difference in skinfold thickness (**a**) and blood pressure (**b**) measures between ART-conceived and spontaneously conceived (SC) children at each study visit, with SC children as the referent. Differences were estimated by multivariable linear regressions adjusted for maternal age, education, ethnicity, household income, height, pre-pregnancy BMI, parity, and any tobacco exposure in the home; paternal age, height, weight, diabetes history, and high blood pressure history; child size, and polygenic risk score for adult adiposity with no multiple testing adjustments. Multiple imputations by chained equations were used to estimate associations while simultaneously accounting for missing covariate values. Skinfold sample sizes (**a**) for each visit (SC/ART): 1038/81; 762/48; 776/54; 827/62; 765/55; 810/63; 779/59; 777/62; 750/55; 743/57. Blood pressure sample sizes (**b**) for each visit (SC/ART): 716/53; 662/44; 701/48; 647/47. Note, not all measures were taken at each visit. Shaded regions connect confidence intervals to aid in visualizing trends and provide no additional information regarding estimates between visits. cm centimeters, m meters, kg kilograms, mmHg milimeters mercury, SDS standard deviation score; colors used to distinguish outcome measures.

Fig. 2), though small numbers ($N = 14$ and 23 ART children, respectively) prevent robust inference. There were no strong associations between ART status and serum cardiometabolic biomarkers at 6 years (Supplemental Table 3 and Supplemental Fig. 2).

Importantly, all relationships remained similar when treatment effects were estimated by doubly robust, collaborative targeted maximum likelihood estimation (C-TMLE) fit by cross-validated, ensemble machine-learning algorithms, with the exception of larger differences in most skinfold thicknesses (except triceps; Supplemental Fig. 3). When emulating an ART trial amongst putatively subfertile couples, differences in weight, BMI, and skinfold thicknesses were less pronounced (Fig. 3); blood pressure differences were almost completely eliminated. Results were consistent across the choice of SC comparison group (Supplemental Fig. 4 shows differences in height $Z$ scores by group). Using propensity score-based (inverse-probability-weighted) models to account for differential follow-up without additional assumptions about missing covariate data gave similar estimates, albeit with less precision (Supplemental Fig. 5).

**Potential pregnancy, intrapartum, and postnatal mediators.** Parental, pregnancy, intrapartum, and infant factors did not explain observed smaller stature and lower blood pressure in ART-conceived children. Specifically, rates of maternal weight gain did not differ by conception status and adjustments for maternal blood pressures in pregnancy, fasting and 2-h OGTT glucose concentrations at 26 weeks gestation, and gestational age at delivery did not affect estimates (Table 2 and Supplemental Table 3). Mode of delivery, number of delivery complications or neonatal diagnoses, and duration of any breastfeeding also did not differ between ART-conceived and SC children. Child hospitalizations, severe diarrhea, fevers, and antibiotic use also did not substantially differ between groups, with ART-conceived children reporting slightly fewer events (Supplemental Table 4).

**Fetal cord tissue DNA methylation.** Under strict Bonferroni correction (187 a priori candidate sites; $P < 2.7 \times 10^{-4}$) and adjusting for maternal age, parity, ethnicity, pre-pregnancy BMI, and child sex, we found ART status to be associated with reduced fetal cord tissue methylation at one site (*cg13403462*;

**Table 2 Child anthropometrics, skinfolds, and blood pressure at 6 years, adjusted for pre-pregnancy and pregnancy factors.**

| | N | Crude β [95% CI] | Adjusted for pre-pregnancy factors[a] β [95% CI] | Adjusted for pre-pregnancy+ pregnancy factors[b] β [95% CI] |
|---|---|---|---|---|
| Height (cm) | 830 | −1.5 (−2.8, −0.2)[c] | −2.4 (−3.6, −1.2)[e] | −2.3 (−3.5, −1.1)[e] |
| Height-for-age Z score (SDS) | 830 | −0.3 (−0.6, −0.1)[c] | −0.5 (−0.7, −0.2)[e] | −0.5 (−0.7, −0.2)[e] |
| Weight (% difference) | 830 | −5.7 (−10.3, −1.0)[c] | −7.9 (−12.2, −3.4)[e] | −7.8 (−12.1, −3.2)[e] |
| Weight-for-age Z score (SDS) | 830 | −0.4 (−0.8, −0.1)[c] | −0.6 (−0.9, −0.3)[e] | −0.6 (−0.9, −0.2)[e] |
| BMI (% difference) | 830 | −3.2 (−6.6, 0.3) | −4.0 (−7.4, −0.5)[c] | −4.0 (−7.4, −0.5)[c] |
| BMI-for-age Z score (SDS) | 830 | −0.3 (−0.7, 0.04) | −0.4 (−0.8, −0.1)[c] | −0.4 (−0.8, −0.03)[c] |
| Skinfold thickness | | | | |
| Triceps (% difference) | 805 | −5.5 (−13.8, 3.6) | −7.4 (−15.5, 1.6) | −7.5 (−15.7, 1.4) |
| Biceps (% difference) | 805 | −7.1 (−15.3, 2.0) | −7.6 (−15.9, 1.5) | −7.6 (−15.9, 1.6) |
| Subscapular (% difference) | 798 | −9.7 (−18.6, 0.2) | −10.4 (−19.2, −0.7) | −10.6 (−19.5, −0.8)[c] |
| Suprailiac (% difference) | 793 | −12.1 (−22.5, −0.4)[c] | −14.2 (−24.3, −2.8)[c] | −14.3 (−24.4, −2.8)[c] |
| Blood pressure | | | | |
| Systolic blood pressure (mm Hg) | 694 | −3.7 (−6.1, −1.2)[d] | −3.3 (−5.9, −0.7)[c] | −2.8 (−5.4, −0.2)[c] |
| Systolic blood pressure, standardized (percentile) | 694 | −10.2 (−16.8, −3.6)[d] | −8.9 (−15.9, −1.8)[c] | −7.6 (−14.7, −0.5)[c] |
| Diastolic blood pressure (mm Hg) | 694 | −2.5 (−4.2, −0.9)[d] | −2.5 (−4.3, −0.7)[d] | −2.1 (−3.9, −0.3)[c] |
| Diastolic blood pressure, standardized (percentile) | 694 | −7.3 (−12.7, −2.0)[d] | −6.3 (−12.0, −0.6)[c] | −5.2 (−11.0, 0.6) |

*cm* centimeters, *SDS* standard deviation score, *mm Hg* milimeters mercury.
[a]Multivariable linear regression adjusted for: maternal age, education, ethnicity, household income, height, pre-pregnancy BMI, parity, and any tobacco smoke exposure in pregnancy; paternal height and weight; child sex, and polygenic risk score for adult adiposity. Multiple imputation by chained equations were used to estimate associations while simultaneously accounting for missing covariate values.
[b]Adjusted for variables in (1) and additionally for: maternal fasting and 2-h post oral glucose tolerance test at 26 weeks of gestation; any resting blood pressure measurements exceeding 140 mmHg systolic or 90 mmHg diastolic at any time during pregnancy; gestational age at delivery; exact age of the child at the visit in days.
[c]$P < 0.05$.
[d]$P < 0.01$.
[e]$P < 0.0018$ (Bonferroni P value 0.05/28 comparisons).

$P = 1.5 \times 10^{-6}$; Supplemental Data) located within the *NECAB3* (N-terminal EF-hand calcium-binding protein 3; i.e., *NIP1*, *ACTL10*) gene body (first exon) on chromosome 20 (Fig. 4). This site was previously identified and replicated in an EWAS of maternal gestational weight gain[16] and is putatively imprinted[15,29]. Moreover, NECAB3 expression has been implicated in upregulating HIF1A- (hypoxia-inducible factor 1-α) activated glycolysis[30], a critical pathway in fertility and pregnancy physiology[17–20].

To follow-up this finding, we further investigated CpGs annotating to *NECAB3* and *HIF* family genes ($N = 94$ additional CpGs). In this exploratory analysis, using a correspondingly revised Bonferroni-corrected threshold of $P < 1.8 \times 10^{-4}$ ($N = 281$), we found stronger associations with reduced methylation at two additional *NECAB3* CpGs: cg14921437 ($P = 1.3 \times 10^{-8}$) and cg03904042 ($P = 2.1 \times 10^{-9}$). Both CpGs surpassed a recommended epigenome-wide threshold[31] of $P < 3.6 \times 10^{-8}$ and cg03904042 replicates a hit from a recent cord blood epigenome-wide association study (EWAS) for ART[24]. Six *HIF* (*HIF1A* and *HIF3A*) CpGs were nominally associated at $P < 0.05$, however, none passed Bonferroni-correction thresholds (Fig. 4). While the original objective was to only follow-up previously reported candidate CpGs, at the reviewer's request we additionally conducted an agnostic EWAS ($N = 336,684$ CpGs; Supplemental Fig. 6) which confirmed the two genome-wide *NECAB3* hits, and suggested three other sites passing epigenome-wide significance (cg09350387 (*FASTKD*), $P = 2.2 \times 10^{-8}$; cg04278794 (*OR2C1*), $P = 5.2 \times 10^{-9}$; and cg22560193 (*APC2*), $P = 3.2 \times 10^{-11}$). For the remaining analyses, we focus on the a priori specified genes and loci.

**Mechanistic simulations.** To investigate whether observed differences in DNA methylation explained associations between conception via ART and child stature and blood pressure, we conducted analyses simulating an ART trial amongst subfertile couples where we were also able to hypothetically manipulate fetal cord tissue DNA methylation. Despite a strong association between ART status and reduced *NECAB3* methylation, we found evidence consistent with a transient and negligible contribution towards mediating attained size (e.g., ~16% of the total effect or 0.05 SD decrease [95% CI: −0.1, −0.004] in weight-for-age at 36 months among ART-conceived children was attributable to cg03904042 methylation differences which declined over time; Fig. 5). When emulating a pragmatic trial, the proportion mediated was further reduced. Results were consistent whether using a single top hit or averaging across *NECAB3* CpGs.

Similar simulation analyses conducted using *HIF* gene methylation found slightly stronger evidence for mediation (Fig. 5). For example, cg27146050 (*HIF3A*) methylation appeared to contribute to ~7% of the lower height observed among ART-conceived children at 6 years (−0.17 cm [−0.3, −0.03] out of a total effect of −2.2 cm [−3.4, −1.1]; 7.7% mediated) in the overall sample. However, this small association diminished when compared against the subfertile cohort (−0.14 cm [−0.5, 0.2] out of a total effect of −2.9 cm [−4.5, −1.3]; 4.8% mediated; Fig. 6). In simulation models testing for residual confounding, we found consistently estimated nulls for maternal *NECAB3* (cg03904042) and *HIF3A* (cg27146050) methylation mediated effects (e.g., −0.0005 cm [−0.23, 0.23] and −0.03 cm [−0.20, 0.13], representing 0.01% and 1% of the total effect of ART on child height at 5.5 years; Supplemental Fig. 7).

**Discussion**
In summary, conception by ART was associated with reduced height and weight, which became more apparent at older ages. Skinfold thickness, fat mass, and blood pressure were also lower. When compared against putatively subfertile couples, as would have been enrolled in a pragmatic trial of ART versus expectant management, blood pressure differences were no longer observed, suggesting residual differences in parental endowment or fertility may explain differences observed in standard analyses. Analyses accounting for various definitions of control populations and accounting for differential loss to follow-up produced consistent results. These associations also persisted when taking into

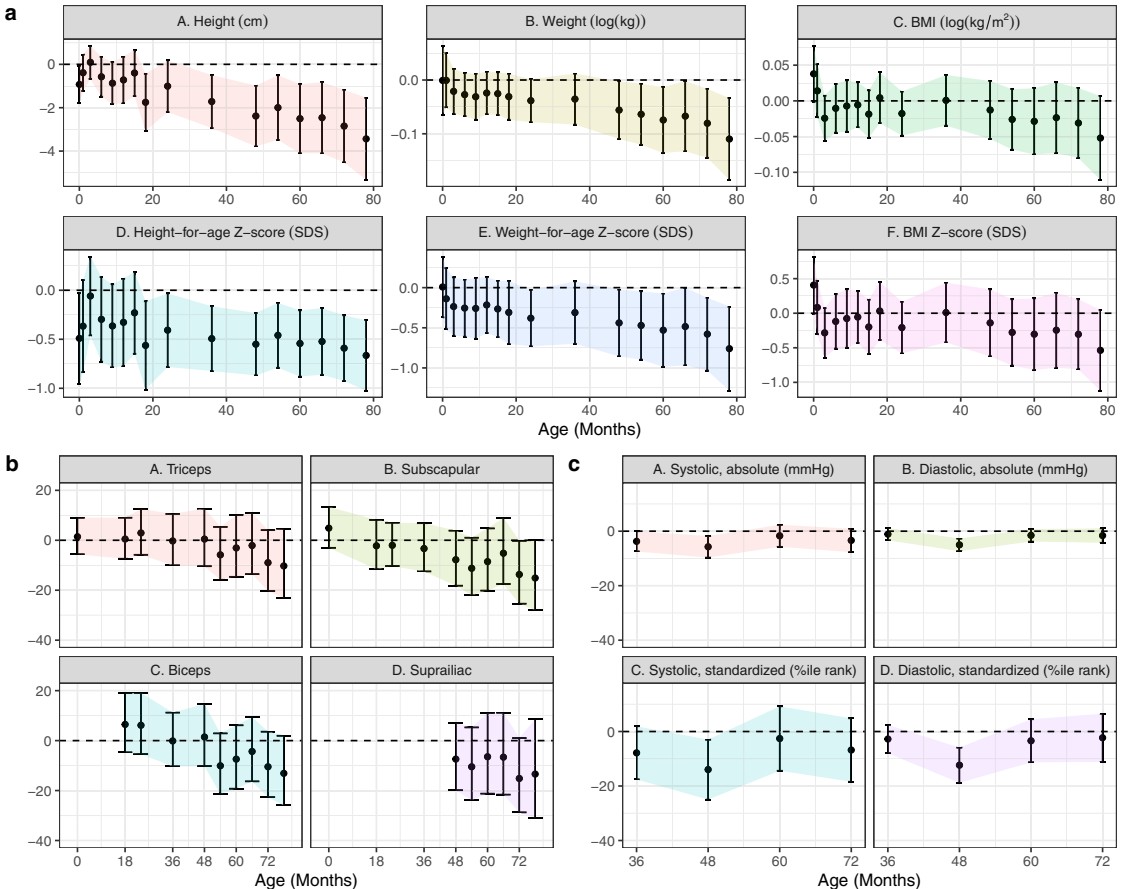

**Fig. 3 Difference in anthropometrics, skinfold thickness, and blood pressure comparing ART-conceived fetuses versus a putatively subfertile cohort, adjusted for pre-pregnancy characteristics.** Point estimates and Wald-type 95% confidence intervals show the adjusted mean difference in anthropometric (**a**), skinfold thickness (**b**), and blood pressure (**c**) measures between ART-conceived and spontaneously conceived (SC) children from a subfertile cohort ($N = 93$) selected on the basis of medical history that may prompt couples to seek fertility specialist care including the history of two or more past miscarriages; medications with potential fertility indications (e.g., aspirin, hormones, thyroid, weight loss); history of PCOS, endometriosis, ovarian cysts, fibroids, or thyroid disorders (hyper- or hypo-). Differences were estimated by multivariable linear regressions adjusted for maternal age, education, ethnicity, household income, height, pre-pregnancy BMI, parity, and any tobacco exposure in the home; paternal height and weight; child size, and polygenic risk score for adult adiposity with no multiple testing adjustments. Multiple imputation by chained equations were used to estimate associations while simultaneously accounting for missing covariate values. Sample sizes for each visit were (SC/ART): (**a**) 93/83; 83/69; 86/71; 84/68; 80/70; 79/68; 79/67; 72/55; 80/60; 84/64; 84/57; 86/64; 80/61; 80/63; 82/59; 77/59; (**b**) 91/81; 69/48; 73/54; 82/62; 81/55; 82/63; 75/59; 77/62; 80/55; 75/57; (**c**) 75/53; 72/44; 70/48; 66/47. Shaded regions connect confidence intervals to aid in visualizing trends and provide no additional information regarding estimates between visits. cm centimeters, m meters, kg kilograms, mmHg milimeters mercury, SDS standard deviation score; colors used to distinguish outcome measures.

account maternal glucose and blood pressures during pregnancy and gestational age at delivery. Other factors which may have bearing on child growth and metabolism, such as fetal growth, maternal gestational weight gain, breastfeeding duration, and child infection and hospitalization history, did not differ. No strong associations were found with other cardiometabolic biomarkers at 6 years.

A recent systematic review[6] of offspring growth found ART-conceived children appeared slightly lighter than spontaneously conceived (SC) children from birth to 4 years, but not at 5+ years, with little evidence of height differences. An earlier systematic review[32] found some evidence for elevated blood pressure and fasting insulin in adolescence and early adulthood, but conflicting evidence for associations with BMI, glucose homeostasis, and dyslipidemia. However, these reviews highlighted critical limitations in past studies including convenience samples of spontaneously conceived children; lack of accounting for parental especially paternal anthropometrics; and inappropriate adjustment for mediators of potential effects such as birth weight.

Notably, parents conceiving via ART are subject to complex selection relative to the general conceiving population, with opposing consequences for child health including higher socioeconomic status and greater financial resources and investments in child development, and adoption and maintenance of health-promoting behaviors under the direction of fertility specialists (e.g., physical activity, weight loss, and smoking cessation). On the other hand, such parents are more likely to be nulliparous, be of advanced parental age, and have underlying endocrine or metabolic disorders (e.g., obesity, hyper-/hypothyroidism, PCOS). Consequently, selection bias and residual confounding when comparing long-term outcomes of ART to spontaneously conceived children is a primary concern that has been insufficiently addressed in prior work. Several studies in which greater attention was given to establishing proper comparison groups and adjusting for familial confounders have found evidence in line with our findings: Kai et al. found that ICSI-IVF-conceived children were 0.3 standard deviations shorter at 3 years compared to SC children enrolled in the same longitudinal cohort (−0.91 vs

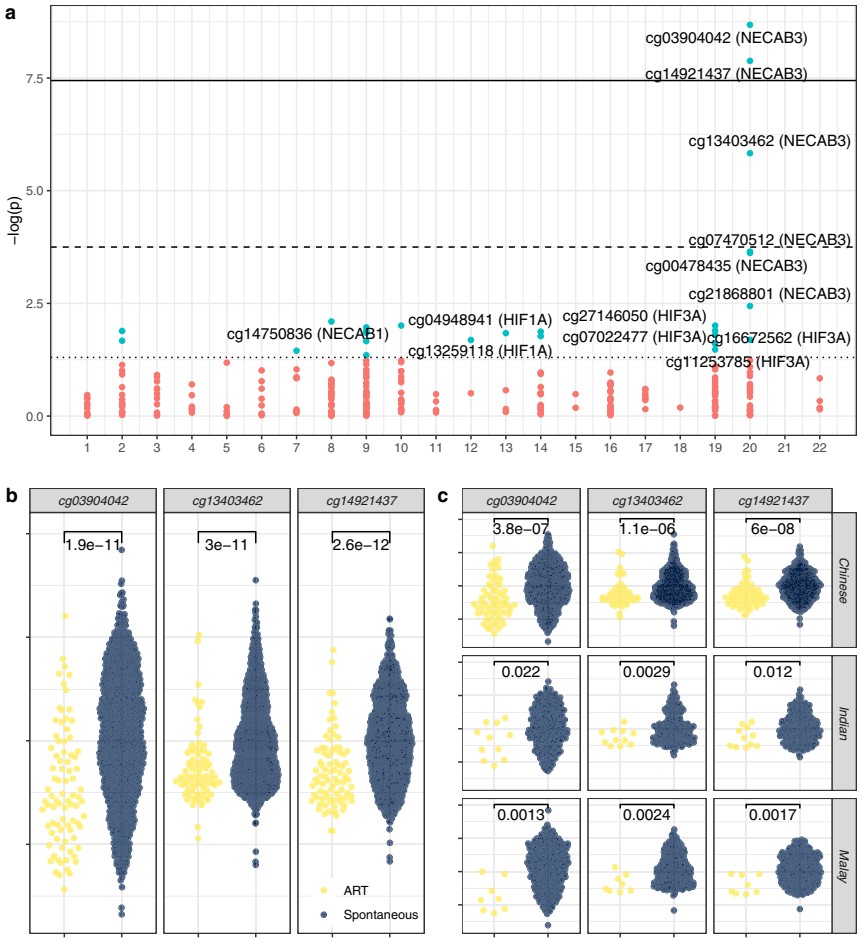

**Fig. 4 Associations between ART status and fetal cord tissue DNA methylation.** Associations between ART status and 281 candidate CpGs (**a**), three top *NECAB3* CpGs (**b**), and three top *NECAB3* CpGs stratified by ethnicity (**c**), were estimated by multivariable linear regression adjusted for maternal ethnicity, age, parity, and pre-pregnancy BMI, and child sex. Horizontal thresholds represent $P = 0.05$; $P = 1.8 \times 10^{-4}$ (Bonferroni correction); $P = 3.6 \times 10^{-8}$ (EWAS-threshold per Saffari et al. 2018[31]). CpGs are grouped by chromosome with colors indicating $P < 0.05$. **b**, **c** show distributions of standardized methylation values with two-sided *t* test *P* values indicated. Sample sizes for all plots = 1094 spontaneous (598 Chinese, 205 Indian, 291 Malay); 83 ART (64 Chinese, 11 Indian, 8 Malay).

−0.61; $P = 0.03$)[33]. At the time of study recruitment, ICSI-IVF was the preferred modality, and 90% of our ART subjects were confirmed to have had ICSI by medical record review. Pontesilli et al. were able to follow children from IVF ($N = 28$) as well as subfertile couples ($N = 220$; defined as conceiving only after 12 months) to 6 years as well as adjust for both maternal and paternal characteristics[34]. They found IVF-conceived children to have lower BMI (−0.9 kg/m²) and blood pressures (−2.1 mmHg systolic and −2.6 mmHg diastolic) when compared to children of subfertile couples at 5–6 years. Unlike our study, however, these analyses were limited by outcome measures at a single time point, comparison cohort formation that did not fully capture potential indications for ART, lacked control for familial confounders such as anthropometric genetic risk scores and postconception factors such as maternal comorbidity and infant breastfeeding, and selection bias due to loss to follow-up.

To address this selection, we formed a spontaneously conceived cohort on the basis of several indicators of subfertility or diagnostic intensity leading to fertility treatment, rather than a single metric such as prolonged time to conception, to mimic inclusion criteria and sampling frame for a hypothetical trial of ART versus expectant management among subfertile couples. We found that using a composite of obstetric and gynecological history,

endocrine dysfunction, and medication record to form the "control arm" served to indirectly balance observed parental characteristics, particularly sociodemographic characteristics suggesting some control for the complex confounding described above. Modifications to the inclusion criteria including specifically paternal risk factors (age > 40, BMI > 35 kg/m², diabetes, or hypertension history) did not change findings based on this approach. One threat to this approach is that follow-up began in early pregnancy (median = 9 weeks) and not prior to conception unlike a true trial[35]. We utilized inverse-probability-weighted models to account for potential selection bias and found results to be consistent. Accounting for differential loss to follow-up from early pregnancy throughout childhood was also shown not to affect estimated differences. Past studies have attempted to address confounding by indication, but not selection bias, using sibling fixed-effects[36] or gestational surrogates[37]. However, such approaches remain confounded by factors that vary between pregnancy and have limited generalizability in utilizing a heavily selected subset of discordant pregnancies[38]. Our approach is more flexible in the selection of comparison populations, importantly including nulliparous women, and incorporate evaluations for sensitivity to comparison populations and some selection biases. Moreover, our anthropometry results were

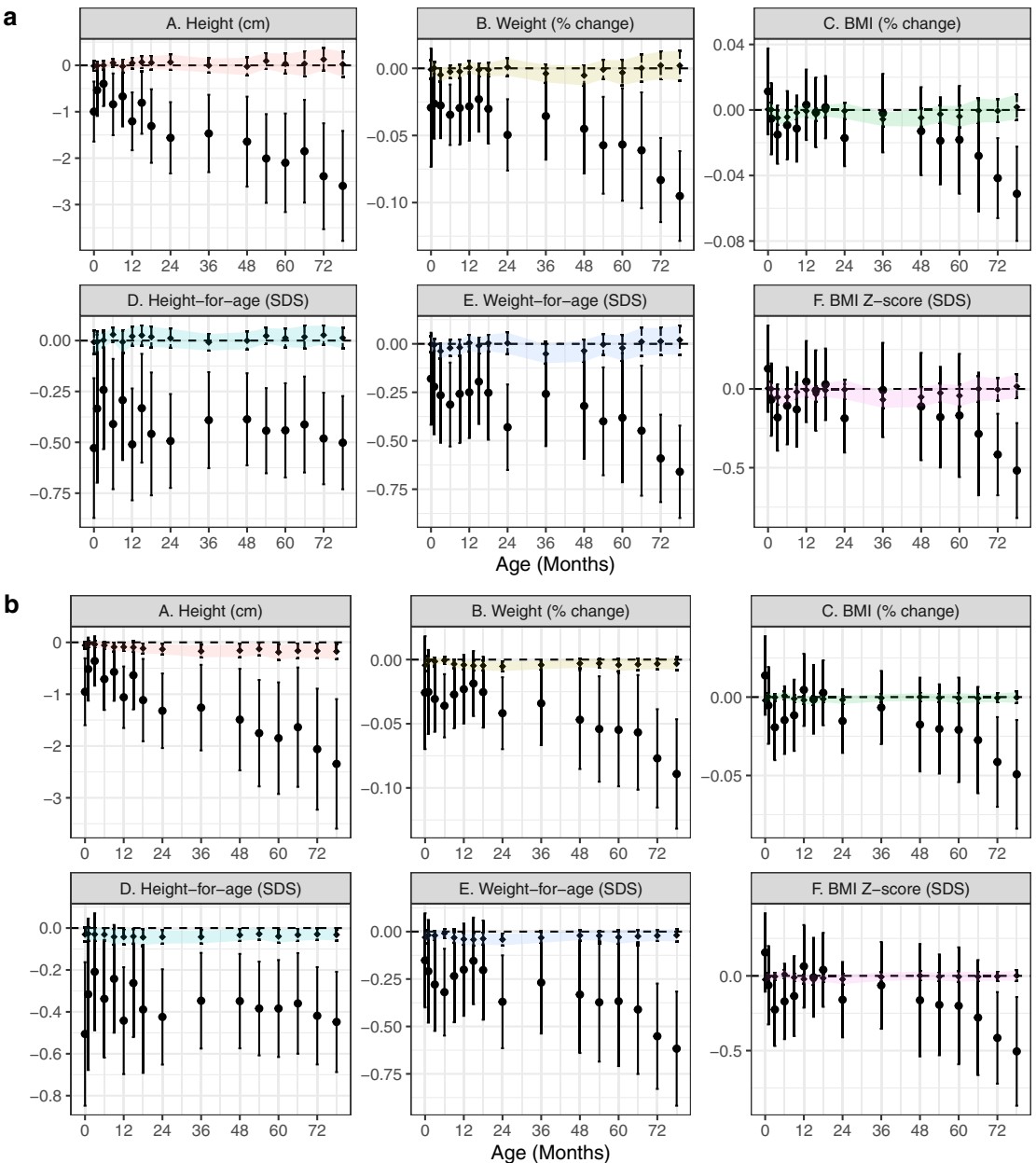

**Fig. 5 Monte Carlo-simulated effects of ART mediated through fetal cord tissue methylation, overall sample.** The solid lines show the estimates and Wald-type 95% confidence intervals for the hypothetical average direct treatment effect of assigning all subjects to ART (versus spontaneous conception) while holding methylation for (**a**) cg03904042 (*NECAB3*) or (**b**) cg27146050 (*HIF3A*) at the untreated (SC) level. Point estimates and standard errors were estimated by parametric g-computation in 100 bootstrapped datasets. The dashed and shaded regions show the specific indirect effect of fixing the methylation level to that caused by ART itself (compared to the level that would have been observed without treatment) among ART-conceived children. Sample sizes across visits are identical to Fig. 1. cm centimeters, SDS standard deviation score; colors used to distinguish outcome measures. Shaded regions connect confidence intervals to aid in visualizing trends and provide no additional information regarding estimates between visits.

robust to a cross-validated machine-learning approach to covariate selection and parameterization as well as weighting and imputation approaches to deal with missing data and loss to follow-up.

Systematic reviews have pointed to the lack of empirical evidence directly linking ART to child phenotypes through putative epigenetic mechanisms[27,39]. Clearance and reestablishment of the embryonic epigenome post-fertilization coincide with assisted reproduction procedures, providing a key window for epigenetic reprogramming, particularly at imprinted genes[11,14,15]. This is supported by the higher prevalence of Beckwith–Wiedemann and Angelman syndromes, resulting from loss of DNA methylation in

imprinting control regions, among ART-conceived children. Risk factors for infertility may also influence nonimprinted pathways. For example, maternal obesity may lead to reduced hypoxia-inducible factor 1 (*HIF1*) expression and thereby poor trophoblast invasion[17], fetal growth[40], and metabolic dysregulation[15]. Increasingly, paternal obesity has also been associated with fetal DNA methylation, including at *HIF* genes[41]. However, such changes may well be transient and the extent to which such variation underlies associations between ART and differential growth, vascular, and metabolic outcomes is unknown. Most associations between pregnancy characteristics and placental or fetal DNA methylation are found to neither persist over time[24]

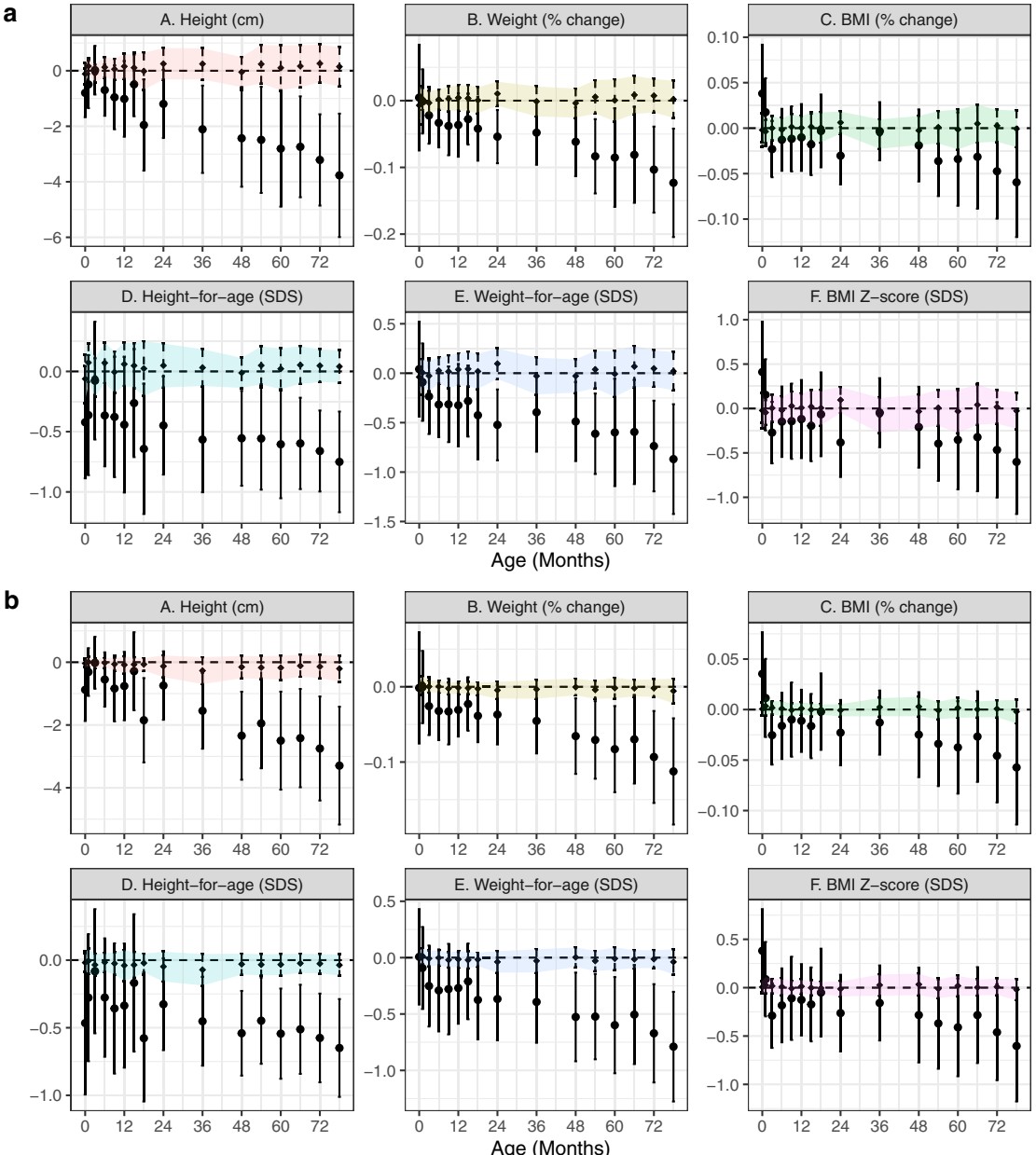

**Fig. 6 Monte Carlo-simulated effects of ART mediated through fetal cord tissue methylation, target trial.** The solid lines show the estimates and Wald-type 95% confidence intervals for the hypothetical average direct treatment effect of assigning all subjects to ART (versus spontaneous conception) while holding methylation for (**a**) cg03904042 (*NECAB3*) or (**b**) cg27146050 (*HIF3A*) at the untreated (SC) level, in an emulated pragmatic trial among subfertile couples. Point estimates and standard errors were estimated by parametric g-computation in 100 bootstrapped datasets. Analyses are similar to those in the overall cohort but restricted to pregnancies from ART ($N = 83$) and spontaneous conceiving couples with infertility indications ($N = 93$). Sample sizes across visits are identical to Fig. 3a. cm centimeters, SDS standard deviation score; colors used to distinguish outcome measures. Shaded regions connect confidence intervals to aid in visualizing trends and provide no additional information regarding estimates between visits.

nor to empirically mediate phenotypes in child or adolescence[16,42]. Rather, observed epigenetic variations may be merely biomarkers for underlying infertility or comorbidities[43,44]. Such observations reinforce the possibility that discovered associations between developmental exposures, including conception via ART, and fetal–placental methylation variation may have limited mechanistic interpretations[45–47]. Recent EWAS has shown conflicting results with respect to ART with some finding associations between ICSI-IVF and placental[46] or cord blood[45] DNA methylation while others suggest differences are explainable by parental characteristics[43]. An important recent study suggests that ART–fetal epigenome associations may not persist into

adulthood[24], and no study has demonstrated whether fetal changes, persistent or otherwise, explain long-term differences in child outcome. Direct empirical evidence for mediation in longitudinal human studies is critically needed.

To this end, we initially investigate associations between ART conception and methylation within the *NECAB3* gene. In a Pregnancy and Childhood Epigenetics (PACE) Consortium meta-analysis ($N = 7523$), Sharp et al. found fetal cord blood methylation at cg13403462 to be negatively associated with maternal BMI (cell-type adjusted $\beta = -1.4 \times 10^{-3}$ per $1 \text{ kg/m}^2$ increased ppBMI) which replicated in adolescent peripheral blood (age ~16–18 years)[16]. In addition, intrauterine causal effects were

supported by a negative control design (methylation associated with maternal but not paternal BMI). We found a similar magnitude of association in our population ($\beta = -1.2 \times 10^{-3}$ per $1\,kg/m^2$ increased ppBMI; $P = 0.083$) after adjusting for maternal age, ethnicity, parity, and sex. A recent GWAS for parent-of-origin effects conducted by Partida et al.[15] leveraging known imprinted regions identified this and other CpGs within *NECAB3* (i.e., *ACTL10*) as putatively imprinted cis-meQTL (methylation quantitative trait loci) proximal to the known MCTS2P pseudogene/HM13 gene (20q11.21) differentially methylated region (DMR)[24]. Most relevantly, a recent study by Novakovic et al.[24] found cg03904042 to be the most strongly associated with ART status in both neonatal and adult blood, a site we replicated as part of our analyses ($\beta = -0.118$; $P$ value $= 8.5 \times 10^{-6}$ vs. $\beta = -0.097$, $P$ value $= 2.1 \times 10^{-9}$, in our study) despite differences in the platform (450 k vs. EPIC) and tissue (fetal cord vs. neonatal blood). Experimental evidence suggests NECAB3 protein depletion suppresses HIF1A (hypoxia-inducible factor) activity[25] which has a critical role in early pregnancy placental vascularization[17,18]. Thus, it is also possible that alterations to *HIF1A* methylation directly may play a role. A past study in this cohort, identified methylation at a separate gene HIF family (*HIF3A*) to be positively associated with birth weight, BMI, and skinfold thickness[48]. We found several CpGs within *HIF1A* and *HIF3A* to be negatively associated with ART status, although not as strongly as *NECAB3*. However, as suggested by Novakovic et al., the relevance of ART-related epigenomic changes to development is likely time-dependent[24]. Consequently, the presence or absence of time point-wise associations, even if repeated, may not be sufficient to suggest functional relevance, and quantitative mediation analyses are necessary. In our subsequent simulation analyses more fully accounting for maternal pregnancy (post conception) characteristics, we found some evidence that cg03904042 (*NECAB3*) and cg27146050 (*HIF3A*) may mediate child weight and height, respectively in a time-dependent manner. Importantly, we found no corresponding evidence of mediation through maternal mid-pregnancy peripheral blood DNA methylation at corresponding sites, supporting the assumptions of no unmeasured common confounders (e.g., maternal genotype or underlying health status) and strengthening the likelihood that estimated fetal epigenome-mediated effects are unbiased. When compared against the subfertile cohort, the mediating roles of *NECAB3* and *HIF3A* were eliminated, further demonstrating the value of a target trial approach in reducing residual confounding.

**Strengths of complementary design and estimation approaches.** To summarize, confidence in our findings was strengthened by triangulating across complementary design and estimation approaches: first, unbiased estimation of ART effects using observational data requires an appropriate selection of comparison groups. We demonstrated our findings were robust to both subgroup restriction and standardization approaches (inverse-probability weighting), with the former less subject to modeling choices and the latter taking better advantage of rich covariate data. Moreover, choosing different subgroup definitions reduced the chance that measurement error for any particular set of measures (e.g., self-report vs. medical records) explained our findings.

Second, we investigated mediating mechanisms to strengthen the evidence that observed ART–phenotype associations are due to proposed mechanisms (e.g., genomic imprinting) and not due to correlated exposures. Mediation analyses come at the cost of stronger assumptions of no unmeasured post-exposure confounding, which we addressed by assessing a negative control

mediator–maternal, mid-pregnancy DNA methylation status. If observed effects are mediated through fetal genomic imprinting alone, it follows that maternal epigenomic variation ought to have no independent effect on the phenotype. However, since maternal epigenome may be affected by ART, direct adjustment for maternal CpGs may result in biased estimates (collider stratification bias). We found no evidence of mediation through corresponding maternal loci which helps to rule out the possibility of either unmeasured common confounders (of both maternal and fetal epigenome) or direct effects of the maternal epigenome.

Third, we found our results were robust to complementary estimation techniques such as mixed effects, which assume random missingness, and multiple imputation, which depend on a proper modeling of the missingness mechanism. Overall, we demonstrated that ART–phenotype associations are robust, and previously observed epigenomic findings may be related to unmeasured differences in underlying maternal/parental health or otherwise noncausal.

Despite our study design and extensive sensitivity analyses, a number of limitations are worth highlighting. While we attempted to mitigate selection bias through inverse-probability-weighted models and alternate comparison groups, the possibility remains that the observed cohorts may not represent those enrolled in a pre-conception study which will need to be addressed in corresponding cohorts. Notably, while we included information on paternal age, anthropometrics, diabetes, and high blood pressure history and showed that differences persisted relative to a comparison arm of children of fathers with higher risk factors, we cannot rule out the possibility that specific unmeasured male infertility factors may underlie observed differences. In addition, measurement error for included covariates or other underlying maternal or familial characteristics not included in our modeling may contribute to residual confounding. Overall, we cannot rule out that better accounting for additional factors may further reduce observed effects. Moreover, on the basis of our study context and inclusion/exclusion criteria (e.g., no type I diabetes mellitus history, willingness to contribute biosamples), our couples may be healthier than others previously studied. Both ART and spontaneous conceiving mothers were generally healthy, with gestational diabetes well-controlled and few women exceeding gestational weight gain guidelines. In addition, we had limited measures of certain child outcomes such as body composition and cardiometabolic biomarkers, so we were unable to assess differences with any precision or over time. Nonetheless, the available measures suggest a consistent phenotype of smaller stature with correspondingly lower total mass and no overt adverse cardiometabolic changes. Finally, we were limited to investigating methylation in maternal peripheral blood (buffy coat) and umbilical cord tissue using the Infinium HumanMethylation450 array. Despite fetal tissues being a natural choice for investigating imprinting errors, we cannot exclude the possibility of important variation in other tissues (e.g., cord blood, placenta) or that variation may occur in genomic CpG sites not covered by the platform. Despite the replication of CpG hits especially cg03904042[24], replication of null mediation findings in independent ART studies will also be important. Moreover, while we found the magnitude of mediation to be generally small and further attenuated in subcohort analyses (<10%), we cannot exclude the possibility that these differences may have been significant at conventional alpha values with larger sample sizes.

The past observation literature examining the influence of assisted reproduction on child growth and cardiovascular health has been hampered by limited follow-up, incomplete consideration of major confounders and comparison groups, and a lack of

empirical evidence for epigenetic mechanisms. Our study builds on the past work by synthesizing complementary approaches to demonstrate that lower height, weight, and skinfold thickness in ART children persist over time and are not explained by major pre- and postnatal factors. Importantly, our target trial approach showed that blood pressure differences may be artefactual due to underlying parental characteristics. Overall, we exhaustively showed that size differences between ART and SC children may be substantial (~0.5 SD for height and weight), but may have no other apparent adverse consequences with respect to blood pressure or cardiometabolic biomarkers. Furthermore, we show several promising biomarkers of ART treatment or infertility did not causally mediate relationships with child stature. In doing so, we present an approach to rule out the potential for direct etiologic links between ART, observed methylation distributions, and child stature. One direction of future work is to further leverage potential tissue-specific effects which may differ by putative pathway: for example to interrogate other imprinted offspring genomic regions using maternal DNA methylation as negative controls and alternatively, to investigate mediation of maternal infertility and comorbidities through maternal and placental epigenomes.

**Conclusions**. Conflicting current evidence on long-term cardiometabolic harms in ART-conceived offspring may influence treatment decisions for those considering assisted reproduction and provoke worry among those who have conceived and delivered through these technologies. Thus, it is critical to produce the best evidence possible given available data and the impracticality of withholding ART completely. Our study clarifies past observational findings by focusing on effects observable in an ideal ICSI-IVF trial while providing further context for the relevance of previously observed epigenetic changes. Our findings support ICSI-IVF producing no adverse early child cardiometabolic outcomes in offspring of subfertile couples despite effects on stature, which should provide some reassurances to providers and families. That said, while our results are internally consistent, they should not be over-interpreted and questions remain. Our study was enabled by the relatively high prevalence of ICSI-IVF treatment which resulted in case numbers comparable to much larger birth cohorts. Nonetheless, as a result of loss-to-follow-up and intensive measures, our effective sample sizes were modest in absolute terms and many exploratory analyses were conducted. Instead, we see these findings as a roadmap for strengthening future analyses, for example in consortia of parent–offspring cohorts with similar genotypic and phenotypic data. Application of these approaches to larger sample sizes, varied population and treatment protocols, and longer follow-up will be necessary for definitive conclusions.

## Methods

**Study setting and conception status ascertainment**. This study was conducted within the "Growing Up in Singapore Towards healthy Outcomes" (GUSTO) prospective birth cohort[49]. Briefly, 1247 women were recruited between June 2009 and September 2010 from women attending first-trimester ultrasound data scans at the two major public maternity units in Singapore: National University Hospital (NUH) and KK Women's and Children's Hospital (KKH). Women were eligible if 18 years and older, Singaporean citizens or permanent residents, with self-reported homogenous ethnic ancestry (Chinese, Indian, Malay), intended to deliver at either of the recruitment hospitals and reside in Singapore for the next 5 years. Women greater than 14 weeks of gestation, receiving chemotherapy, psychotropic medications, or having an existing type I diabetes mellitus diagnosis at the time of recruitment were excluded. Women who ultimately did not agree to donate birth tissues (cord, placenta, cord blood) were also excluded. Women were asked to self-report whether the current pregnancy was conceived via IVF and the use of assisted reproductive technologies was confirmed via medical record review by a senior obstetrician and fertility consultant. Women reporting IVF conception with

multiple gestations were further excluded. Informed consent was provided by all mothers. For this study, five spontaneously conceived (SC) twin pairs were also excluded resulting in a base study population of 1237 ($N = 85$ ART; 1152 SC). Study protocols following the principles of the Declaration of Helsinki and were approved by the respective ethics committees for the two hospitals: National Healthcare Group Domain Specific Review Board (NUH) and SingHealth Centralized Institutional Review Board (KKH). All participants included in this study provided informed consent to participate and contribute their data to publications. The GUSTO cohort study is registered under study ID: NCT01174875 (clinicaltrials.gov) which broadly covers investigations of parental and gestational influences on child health. The specific analyses conducted here were not pre-specified in the protocols.

**Parental covariate information**. Maternal obstetric and medical history including self-reported pre-pregnancy body weight, sociodemographic characteristics, and health behaviors, such as personal and family tobacco smoking, were ascertained by study staff administered standardized questionnaire at recruitment and at a study visit at 26–28 weeks gestation. Self-reported pre-pregnancy weight is preferred overweight in pregnancy as interim weight changes may be causally related to the probability of successful conception resulting in bias. In any event, self-reported weight was highly correlated with weight at booking (Pearson's $r = 0.97$). At the 26–28-week visit, anthropometrics including height, weight, skinfold thickness, and mid-upper arm and waist and hip circumferences were taken by study staff using standardized protocols and a 75-g oral glucose tolerance test (OGTT) administered with blood draws for fasting and 2-h post-OGTT glucose. Blood samples were immediately spun and plasma glucose was assayed by the hexokinase enzymatic method at the respective hospital laboratory at which it was drawn (Abbott Architect c8000; Abbott Manufacturing; Irving, USA and Beckman AU5800; Beckman Coulter, Brea, USA, at KHH and NUH, respectively). Antenatal blood pressures, diagnoses, and medications were abstracted from clinical case notes. Father's height and weight were measured by study staff at 24 months (or 36 months if they did not attend the 24-month visit) using identical protocols as for maternal measures. As with maternal measures, postconception paternal weight measures may result in biased estimates. Quantitative bias analysis investigating the potential for differential measurement error (i.e., 24-month weight as a proxy for pre-conception weight) show that even extremely unbalanced error (e.g., fathers of ART children losing an average of 20 kg more than SC children between pre-conception and 24 months) would minimally impact estimates (−0.45 [95%CI: −0.69, −0.21] to −0.42 [−0.69, −0.16], respectively).

**Fetal and child growth parameters**. Fetal gestational age and estimated date of delivery were calculated based on dating ultrasound. Fetal biparietal diameter (BPD), head (HC) and abdominal (AC) circumferences and femur and humerus lengths were measured by trained sonographers at 19–21, 26–28, and 32–34 weeks. Estimated fetal weights (EFW) were calculated using the Shepard formula as follows: $EFW = 10\char`^(−1.7492 + 0.166*BPD + 0.046*AC − 2.646*(AC*BPD)/1000)$, where BPD and AC are expressed in centimeters and EFW in kilograms. Mode of delivery, procedures, and complications and birth weight, length, and head circumference were abstracted from the delivery records. At all postdelivery visits weight (calibrated Seca 334 or Seca 803 digital scales; Seca, Hamburg, Germany); recumbent crown-to-sole length (up to 24 months; Seca 210 Mobile Measuring Mat) / standing height (beginning at 18 months; Seca 213 Stadiometer); head, mid-upper arm, and abdominal circumferences (inelastic measuring tape); and skinfold (triceps, biceps, subscapular, and suprailiac) thickness (Holtain skinfold calipers; Holtain Ltd., Crymych, UK) were collected by trained study staff in duplicate or triplicate (or 4–5 time for skinfold) and averaged under standardized protocols based on U.S. National Health and Nutrition Examination Survey (NHANES) protocols. Age- and sex- standardized length/height-for-age, weight-for-age, and BMI-for-age Z-scores for based on WHO Multicentre Growth Reference Study distributions as implemented in the *igrowup* Stata macro[50,51]. Because descriptive findings showed that length distributions were identical for children with and without height measures at 18 and 24 months, length was used for calculations unless only height was available, in while case the standard correction (height + 0.7 cm) was applied. Exact age in days was used for all calculations. Measures from 16 visits were used in this study: 0 (birth), 1 (~3 weeks), 3, 6, 9, 12, 15, 18, 24, 36, 48, 54, 60, 66, 72, and 78 months.

**Child blood pressures, body composition, and serum cardiometabolic biomarkers**. Child blood pressures were measured at 3-, 4-, 5-, and 6-year visits using a DINAMAP CARESCAPE V100 (GE Healthcare, Milwaukee, USA) automated blood pressure monitor. Children were fitted with an appropriate sized cuff (8–13 cm or 12–19 cm; GE CRITIKON) on the bare, upper arm and, after a 5-min initial rest in the seated position, measured twice with 25–30 s between measurements. If either systolic or diastolic blood pressure varied more than 10 mmHg, a third measure was taken. A simple mean was taken of all valid, repeated measures for a subject. Age-, sex-, and height-standardized Z scores for systolic and diastolic blood pressure were calculated based on American Academy of Pediatrics clinical practice guidelines (Flynn et al.)[52] and as implemented in a Stata macro by

Sørensen and Bruun[53]. At ages 5 and 6, total lean and fat mass (in kilograms) was estimated (Chen et al. 2018) in a subset of children by quantitative magnetic resonance (EchoMRI-Adolescent Humans Body Composition Analyzer; EchoMRI Corporation, Singapore) under a low magnetic field (0.007 Tesla). At age 6, children were asked to fast the evening before the study visit. At the visit, venepuncture was performed by study staff and a peripheral blood sample was spun, aliquoted, and stored at −80 °C. One plasma aliquot was immediately assayed for glucose concentrations as reported above for maternal glucose. In 2019, one serum aliquot per child was thawed and analyzed for the following biomarkers at the College of American Pathologist-accredited NUH Referral Laboratory (Singapore) following standard clinical laboratory protocols: AST, ALT, high-sensitivity C-reactive protein (hsCRP), creatinine, total cholesterol, triglycerides, high- (HDL) and low- (LDL) density lipoprotein-cholesterol, GGT (all Beckman AU58000), and insulin (Beckman DXL800). Fasting glucose (mmol/L) and insulin (mIU/L) were used to calculate a homeostatic model of assessment measures of percent beta-cell function (HOMA-β = [20*insulin]/[glucose − 3.5]) and inverse of percent insulin sensitivity, or insulin resistance (HOMA-IR = [insulin*glucose]/22.5). HOMA-β for children with fasting glucose ≤3.5 mmol/L ($N = 5$ spontaneous conceptions) were set to missing.

**Child postnatal health and behaviors**. Whether infants were continuing to exclusively or partially breastfeed was reported by mothers at follow-up visits up through 12 months. Thus, breastfeeding can be categorized into intervals with a lower bound being the most recent visit in which mothers confirmed children were still actively breastfeeding. Mothers were also asked about her child's general health in the preceding 3 months at all visits. These include frequency of diarrhea lasting 2 or more days, number of fevers exceeding 38 °C, number of hospitalization, and number of days of antibiotic use, if any.

**Child polygenic risk score**. A polygenic risk score for adiposity previously constructed in this study[54] was included to capture the potential contributions of genetic ancestry to infertility and child stature. Briefly, child genotyping was conducted via Illumina Omniexpress + exome array and called via standard protocols in GenomeStudio Genotyping Module and GenCall v1.8 (Illumina, San Diego) 577,204 SNPs were retained after removal of non-autosomal SNPs, call rates <95%, minor allele frequencies <5%, and sites not in Hardy–Weinberg Equilibrium. Polygenic risk scores (PRS) were computed based on coefficients and $P$ values for SNPs significantly associated with adult BMI in the GIANT (Genetic Investigation of Anthropometric Traits) consortium[55]. Two rounds of clumping were performed in PLINK 1.9:[56] First, SNPs with the smallest GIANT-reported $P$ value in each 250-kb window were selected and SNPs with $R^2 > 0.5$ were removed. Second, a 5-Mb window was used with a $R^2 > 0.2$ thresholds. Individual PRS were computed by summing the retained alleles, weighted by the GIANT-reported regression coefficient. This was conducted separately for each of the three self-reported ethnicities and alternative PRSs were generated by repeating the procedure across a range of $P$ value filtering thresholds ($10^{-10}$–1). Within ethnicities, each PRS was standardized to mean 0 and variance 1. For each ethnicity, the $P$ value threshold producing a PRS with the strongest correlation to child birth weight and BMI was retained (0.5, 0.1, and $10^{-4}$, for Chinese, Malay, and Indian, respectively).

**Fetal cord tissue DNA methylation**. Fetal cord tissue was cleaned, segmented, and stored at −80 °C until DNA extraction. Standard protocol for quantification via Infinium HumanMethylation450 array were followed and raw $β$ values were exported from GenomeStudio Methylation Module v1.8 (Illumina, San Diego). Probes with <3 beads, detection $P > 0.01$, cross-hybridizing, SNPs, or on sex chromosomes were be excluded. Principle components of $β$ values were regressed against technical covariates including lab technician, batch, plate, and location, and batch effects corrected via an Empirical Bayes algorithm (ComBat). Reference-free algorithms were used to estimate cell-type proportions. Finally, back-converted $β$ values will be regressed on indicators for chip position, DNA extraction, and bisulfite conversion batch and residuals retained. For the purposes of this study, we matched 187 a priori CpGs (out of a total 224 searched) previously shown in the literature to be associated in offspring tissues with child size or, importantly, maternal factors only if demonstrated in a longitudinal setting. CpGs were selected either based on exact matches from EWAS conducted on the same platform, or within + /−2 kb of locations from candidate gene studies. After our initial findings, we added an additional 94 CpGs (out of 113 searched) annotating to the *NECAB3* and *HIF* family (*HIF3A*, *HIF1A*) genes. Both sets and the referring studies are shown in Supplemental Data.

**Maternal 26-gestational week buffy coat DNA methylation**. Maternal fasting blood ($N = 915$) was collected at 26-gestational week visit and immediately fractionated and stored at −80 °C until DNA extraction. Standard protocol for quantification via Infinium MethylationEPIC array were followed and processing and QC pipelines mirrored protocols for fetal cord tissue. For the purposes of the study, chip position, DNA extraction, and bisulfite conversion batch-adjusted residuals for only three probes were used: cg03904042 (*NECAB3*), cg27431396 (*NECAB3*), and cg27146050 (*HIF3A*).

## Statistical analyses

*Data description*. We describe all variables by plotting probability density functions and histograms and univariable statistics including means/standard deviations, medians/interquartile ranges, or proportions/counts, as well as proportions of missingness for all variables at all visits. Differential subject retention and availability of outcome measures was tabulated and plotted. We found little evidence for differences in retention, with slightly higher overall participation among the ART group, or in outcome distributions of individuals eventually lost to follow-up (Supplemental Fig. 8).

*ART–child outcome associations*. To investigate the associations between ART status and child outcomes, we followed four general approaches:

First, for each outcome measure, we fit multivariable linear regression models predicting the respective outcome by ART status and outcome predictors that precede conception (classic, time-invariant confounders): maternal age, education, ethnicity, household income, height, pre-pregnancy BMI, parity, and any tobacco smoke exposure in pregnancy as a proxy for peri-conception smoke exposure; paternal age, height, weight, diabetes history, and high blood pressure history; child sex, and polygenic risk score. In addition, because they may capture additional variability arising from parental comorbidity despite plausibly existing on the causal pathway, we estimate models further adjusting for maternal fasting and 2-h post oral glucose tolerance test at 26 weeks of gestation; any resting blood pressure measurements exceeding 140 mmHg systolic or 90 mmHg diastolic at any time during pregnancy; gestational age at delivery; and as a precision variable, the exact age of child at visit in days. For all models, multiple imputations by chained equations were used to stochastically impute ten datasets with complete covariate information. Point estimates for each association were derived by averaging across imputed sets and 95% confidence intervals were calculated by Rubin's rules for calculating standard errors from within- and between-imputation standard deviations.

Second, for outcomes for which we had repeated measures (height, weight, BMI, blood pressure, and skinfold thicknesses), we utilized linear mixed models to account for within-person variation in outcomes under the strong assumption that visits are missing at random. To do this, we estimated a multilevel model of a similar form as the main models (same set of time-invariant confounders), but allowing for random slopes and intercepts within subjects using an unstructured covariance matrix and introducing an interaction term between visit and ART status. Results from these analyses were qualitatively similar (i.e., lower height, weight, and blood pressures), so only the single-level multivariable estimates are presented for consistency across outcomes. Plots of mean group-level anthropometric measures predicted by multilevel models demonstrate these consistent findings (Supplemental Fig. 9).

Third, to account for differential selection into the study and observations across time points, we estimated our models weighted for the inverse probability of ART treatment and censorship. We did this by using multivariable logistic regression to separately predict the probability of (1) ART status by time-variant covariates and (2) being censored at birth or each subsequent visit by ART status and time-invariant covariates. For each time point, an inverse-probability weight was generated for each observation as the product of the inverse predicted probability of treatment and the inverse probability of censorship. Then each main multivariable linear regression was fitted using these weights. These models were estimated in a complete case setting to contrast with our main multiply imputed results, which trades off some statistical power by allowing different assumptions regarding covariate missingness—namely that there is no differential missingness conditional on the probability of observation at the subject (rather than covariate) level. In this way, any differential dropout, including pregnancy loss, will be standardized across treatment groups.

**Doubly robust ART effect estimation with cross-validated, machine learning**. Fourth, because both outcome modeling (approach one; covariate adjustment) and treatment modeling (approach three; propensity score weighting) approaches may be subject to misspecification of either the covariate set or functional forms, we implemented targeted maximum likelihood estimation (TMLE), which is doubly robust in that the estimator is unbiased if either model is correct and bias decreases as a function of the bias of both outcome and treatment models, otherwise[57]. Importantly, the use of such doubly robust estimators enables the implementation of semi- and non-parametric machine-learning estimators[58], whereas their usage in standard ("singly robust") outcome regression and propensity score models may in fact lead to greater bias[59]. For this study, Collaborative-TMLE (C-TMLE; ctmle package in R) was implemented to prevent overfitting in the setting of sparse treatment models[60].

For each model, an initial outcome regression was fit using the same ART exposure and covariates as approach one (multivariable regression), but with outcome Y scaled to [0,1]. An intercept-free logistic regression was fit repeatedly on Y using the predicted value from step 1 as an offset and an iteratively-updated propensity score as the only covariate. The propensity score was iterated by sequentially adding covariates to minimize the error in predicting Y. Once a minimum is found, the average treatment effect (ATE) was computed by taking the inverse-logit (expit) of the sole parameter of this final (influence function) regression and back-converting to the original scale. Standard errors (SE) were

calculated from the variance of this influence function and Wald-based 95% confidence intervals constructed by ATE $+/- 1.96 *$ SE.

Notably, this estimator can be used with standard regression approaches for the initial outcome and propensity score estimations without employing non-parametric algorithms (machine learning). However, as mentioned above, a key strength of this estimator is the ability to incorporate machine learning (coupled with cross-validation to prevent overfitting) to reduce residual confounding in both the outcome or treatment models. To this end, all outcome and propensity score predictions were estimated via SuperLearner (a clear introduction is given by Naimi & Balzer:[61] For each model, the data were split into five equal parts with 4 used for fitting a predictive model (training) and 1 held out for estimating fitted values (testing). A library of algorithms was fit to the training data including GLM, GLM with interactions, Bayes GLM, observed means, neural net with a single hidden layer, and XG Boost (boosted trees). For each algorithm, fitted values are computed five times, each time taking a different combination of four training parts and the 5th test set, and then averaged. This results in a set of cross-validated predicted values for each algorithm. To compute the contribution of each algorithm to the final model, a model (non-negative least squares) is fit regressing the observed outcome by the predictions of each algorithm. The coefficients for this fit then become weights for the prediction of each algorithm with better-performing algorithms effectively contributing more to the prediction. Finally, the set of algorithms are fit to the original dataset and their fitted values are weighted by these estimated coefficients to form the final best prediction.

**Target trial comparison cohort formation.** A concern with prevailing studies is that samples of spontaneously conceived children drawn from pediatric clinic visitors or volunteers from the general population are inappropriate comparators to children conceived via ART. Moreover, the selection bias introduced by such convenience samples can be conflicting since couples seeking ART services may simultaneously have more comorbidities (e.g., diabetes mellitus, obesity), but be of higher socioeconomic status and have adopted risk-reducing behaviors such as smoking cessation. In addition, spontaneously conceiving healthy couples enrolled in early pregnancy would have drastically different fertility and conception experiences than ART-conceiving couples enrolled at the same relative time. To address this, we "enrolled" a control group to more closely represent the population from which the ART pregnancies were drawn, that is, couples who may have sought fertility treatment and been eligible for enrollment in an RCT of ART versus expectant management (if such a trial were ethically feasible) by the nature of a priori medical history of risk factors for infertility and medication and diagnoses indicating a higher probability of attending obstetric and specialty care. Specifically, we selected all spontaneously conceived children whose mother reported >1 previous miscarriage, PCOS, uterine fibroids, ovarian cysts, or hyper-/hypothyroidism diagnosed prior to the current pregnancy or medications indicative of treatment for endocrine disorders or fertility difficulties (e.g. aspirin, duromine, progesterone, clomifene, cabergoline, carbimazole, dexamethasone, dydrogesterone, prednisolone, propylthiouracil, thyroxine, thiamazole).

This resulted in a comparison group of 93 spontaneously conceived children who were more similar to ART-conceived children in both measured sociodemographic characteristics and medical history, and ideally, other unmeasured characteristics related to treatment-seeking, fertility status, and child health. Moreover, we note the putatively subfertile cohort attended the first ultrasound visit at an earlier early median gestational age than the general spontaneously conceiving couples group (8.1 weeks vs. 9 weeks, respectively), suggesting they were followed more intensely than the typical healthy couple. In other words, this approach attempts to emulate a target randomized clinical trial[26] of expectant management versus ART among subfertile couples planning pregnancy using nonrandomized data. All above models were fit restricted to the 83 ART-conceived and 93 spontaneously conceived subcohort. Notably, this primary definition did not include any paternal risk factors because paternal health information was limited to a potentially non-random subset of fathers and we did not want imputation decision to affect this estimation strategy. Nonetheless, male infertility was noted in a majority of medical records in the ART cases, thus important to balance in comparison cohorts. We note that our primary cohort was able to indirectly balance paternal weight and high blood pressure (Supplemental Table 1), nonetheless, we formed alternative comparison arms based on the following paternal risk factors: age >40 at first birth; BMI > 35; any diabetes diagnosis prior to study enrollment; and any high blood pressure diagnosis prior to study enrollment. Subsequently, we also estimated models comparing ART children ($N = 83$) against three additional comparison cohorts as follows: (1) the original definition plus any paternal risk factors ($N = 200$); (2) any paternal risk factors only ($N = 121$); and (3) a random draw of all individuals with none of the indications ($N = 204$).

**Fetal cord tissues associations.** We investigated the association between ART status and fetal cord tissue methylation by fitting separate multivariable linear regression predicting each of 187 a priori CpG sites by ART status, maternal ethnicity, age at delivery, parity, and pre-pregnancy BMI, and child sex. A strict Bonferroni correction ($P < 0.05/187 = 2.7 \times 10^{-4}$) was applied for initial discovery. Additional adjustments for polygenic risk score did not alter results. From this result, 94 additional CpGs annotating to *NECAB3* or *HIF3A* or *HIF1A* were added

to the analysis, and the Bonferroni correction was modified accordingly ($P < 0.05/281 = 1.8 \times 10^{-4}$). Q–Q plots did not show appreciable $P$ value inflation.

**Mechanistic simulation.** We estimated the potential mediating effects of significantly association CpGs in the relationship between ART status and child stature using parametric g-computation by Monte Carlo simulation (bootstrapping). Briefly, we fit a system of multivariable linear equations was fit based a priori specified causal relationships between ART conception, maternal fasting glucose (FG), systolic blood pressure in pregnancy (SBP), gestational age at delivery (GA), fetal cord tissue methylation at a given site (CPG), child anthropometrics (Y), and confounders. These equations take the following forms:

$$FG = \beta_1 * ART + \Gamma_1 * \text{confounders} + \alpha_1 \qquad (1)$$

$$SBP = \beta_2 * ART + \beta_3 * FG + \Gamma_2 * \text{confounders} + \alpha_2 \qquad (2)$$

$$GA = \beta_4 * ART + \beta_5 * FG + \beta_6 * SBP + \Gamma_3 * \text{confounders} + \alpha_3 \qquad (3)$$

$$CPG = \beta_7 * ART + \beta_8 * FG + \beta_9 * SBP + \beta_{10} * GA + \Gamma_4 * \text{confounders} + \alpha_4 \qquad (4)$$

$$Y = \beta_{11} * ART + \beta_{12} * FG + \beta_{13} * SBP + \beta_{14} * GA + \beta_{15} * CPG + \Gamma_5 * \text{confounders} + \alpha_5 \qquad (5)$$

Where $\Gamma$ represent vectors of coefficients for the time-invariant confounders: maternal age, education, ethnicity, parity, height, pre-pregnancy BMI, and smoking exposure in pregnancy; household income, paternal height, paternal weight, child sex, and polygenic risk score.

Observed data were resampled with replacement 100 times. In each set, parameters were estimated for this system of equations, and anthropometric measurements were estimated using observed covariates and alternatively assigning either 1 or 0 to ART status. The total effect (TE) is computed as the mean difference between assigning all individuals to ART and non-ART. To estimate natural direct effects (NDE) independent of methylation level, the same computations are performed for ART = 0, however, for ART = 1, methylation level is counterfactually set to the level predicted by ART = 0. The estimate for the natural indirect effect is taken as the difference TE – NDE, or equivalently estimated as the difference when setting methylation to the level observed under ART = 1 versus that observed under ART = 0 among those for whom ART = 1. To estimate variability, the standard deviation across bootstrap samples is taken as the standard error and normal-based 95% confidence intervals computed (percentile-based intervals did not differ substantially). G-computation was performed on both the full dataset and restricted to the subcohort ("target trial"). There were minimal differences between models so the target trial results are reported.

**Negative control analyses.** Under a putative fetal genomic imprinting hypothesis, maternal epigenome ought to have no direct effects on offspring phenotype when pregnancy comorbidities are accounted for (Supplemental Fig. 10). Thus, if maternal CpGs mediated effect is detected independent of maternal complications this indicates either: residual confounding by unmeasured parental characteristics (estimates for fetal epigenetic mediation are biased) or there is a direct effect of maternal epigenome (fetal imprinting hypothesis is inadequate). To evaluate this, we recompute the mechanistic simulations using corresponding maternal mid-pregnancy methylation residuals in place of the fetal cord tissue measures. Moreover, to test the sensitivity to the temporal and causal ordering of the indicated diagram, we also compute the mediated effect after reordering our system of equations as follows (where $CPG_M$ indicate the maternal methylation value):

$$CPG_M = \beta_1 * ART + \Gamma_1 * \text{confounders} + \alpha_1 \qquad (6)$$

$$FG = \beta_2 * ART + \beta_3 * CPG_M + \Gamma_2 * \text{confounders} + \alpha_2 \qquad (7)$$

$$SBP \beta_4 * ART \beta_5 * CPG_M + \beta_6 * FG + \Gamma_3 * \text{confounders} + \alpha_3 \qquad (8)$$

$$GA = \beta_7 * ART + \beta_8 * CPG_M + \beta_9 * FG + \beta_{10} * SBP + \Gamma_4 * \text{confounders} + \alpha_4 \qquad (9)$$

$$Y = \beta_{11} * ART + \beta_{12} * CPG_M + \beta_{13} * FG + \beta_{14} * SBP + \beta_{15} * GA + \Gamma_5 * \text{confounders} + \alpha_5 \qquad (10)$$

The computed natural indirect effect (NIE) therefore should correspond to zero if both of our assumptions about unmeasured confounding and no-direct effect of maternal CpGs is correct and non-zero if either are violated. We found fairly precise estimates of null mediation by maternal CpGs which was invariant to causal ordering, strengthening our confidence that our estimates of fetal cord tissue mediation are unbiased if small and potentially underpowered.

All data processing, analyses, and visualizations for this study were conducted in either Stata 15.1 SE (StataCorp, College Station, Texas) using base and gformula macros or RStudio 1.2.1335 (RStudio, Inc., Boston, Massachusetts)/R 3.6.0 (R Core Team, Vienna Austria) using the stats 3.6.0, sva 3.22, minfi 1.22.1, haven 2.3.1, tidyverse 1.3.1, ggplot2 3.3.2, readxl 1.3.1, ggbeeswarm 0.6.0, ggrepel 0.9.0; ggpubr 0.4.0, ctmle 0.1.2, and SuperLearner 2.0.26 packages as well as their respective dependencies.

**Reporting summary**. Further information on research design is available in the Nature Research Reporting Summary linked to this article.

## Data availability

Methylome data and basic participant characteristics are deposited with the NCBI Gene Expression Omnibus under Series accession number GSE158064. An anonymized dataset and raw results are available through github (github.com/jhuang35/ivf_growth/) and Zenodo (https://doi.org/10.5281/zenodo.4662336)[62]. Other data are available upon reasonable request to the GUSTO study executive committee. A working overview of available data and variables can be found at sicsdatavault.sg/gusto/. The corresponding author may be contacted for further details.

## Code availability

Custom analytic code and code to reproduce tables and figures are available on github (github.com/jhuang35/ivf_growth/) and Zenodo (https://doi.org/10.5281/zenodo.4662336)[51].

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

## Acknowledgements

The authors would like to thank the participants of the GUSTO study for their unwavering enthusiasm over the years and the diligent, meticulous work of affiliated research staff, students, and postdocs, without whom the research would be impossible. This research is supported by the Singapore National Research Foundation under its Translational and Clinical Research (TCR) Flagship Programme and administered by the Singapore Ministry of Health's National Medical Research Council (NMRC), Singapore —NMRC/TCR/004-NUS/2008; NMRC/TCR/012-NUHS/2014. This research is also supported by NMRC/OFYIRG19Nov-0033 to J.Y.H. K.M.G. is supported by the UK Medical Research Council (MC_UU_12011/4), the National Institute for Health Research (NIHR Senior Investigator (NF-SI-0515-10042) and NIHR Southampton Biomedical Research Centre (IS-BRC-1215-20004)), the European Union (Erasmus + Programme ImpENSA 598488-EPP-1-2018-1-DE-EPPKA2-CBHE-JP) and the British Heart Foundation (RG/15/17/3174). Additional funding is provided by the Singapore Institute for Clinical Sciences, Agency for Science Technology and Research (A*STAR), Singapore. The GUSTO study group includes Allan Sheppard, Amutha Chinnadurai, Anne Eng Neo Goh, Anne Rifkin-Graboi, Anqi Qiu, Arijit Biswas, Bee Wah Lee, Birit F.P. Broekman, Boon Long Quah, Borys Shuter, Chai Kiat Chng, Cheryl Ngo, Choon Looi Bong, Christiani Jeyakumar Henry, Claudia Chi, Cornelia Yin Ing Chee, Yam Thiam Daniel Goh, Doris Fok, E Shyong Tai, Elaine Tham, Elaine Quah Phaik Ling, Evelyn Chung Ning Law, Evelyn Xiu Ling Loo, Fabian Yap, Falk Mueller-Riemenschneider, George Seow Heong Yeo, Helen Chen, Heng Hao Tan, Hugo P. S. van Bever, Iliana Magiati, Inez Bik Yun Wong, Ivy Yee-Man Lau, Izzuddin Bin Mohd Aris, Jeevesh Kapur, Jenny L. Richmond, Jerry Kok Yen Chan, Joanna D. Holbrook, Joanne Yoong, Joao N. Ferreira., Jonathan Tze Liang Choo, Jonathan Y. Bernard, Joshua J. Gooley, Keith M. Godfrey, Kenneth Kwek, Kok Hian Tan, Krishnamoorthy Niduvaje, Kuan Jin Lee, Leher Singh, Lieng Hsi Ling, Lin Lin Su, Ling-Wei Chen, Lourdes Mary Daniel, Lynette P. Shek, Marielle V. Fortier, Mark Hanson, Mary Foong-Fong Chong, Mary Rauff, Mei Chien Chua, Melvin Khee-Shing Leow, Michael Meaney, Mya Thway Tint, Neerja Karnani, Ngee Lek, Oon Hoe Teoh, P.C. Wong, Paulin Tay Straughan, Peter D. Gluckman, Pratibha Agarwal, Queenie Ling Jun Li, Rob M. van Dam, Salome A. Rebello, Seang-Mei Saw, See Ling Loy, S. Sendhil Velan, Seng Bin Ang, Shang Chee Chong, Sharon Ng, Shiao-Yng Chan, Shirong Cai, Shu-E Soh, Sok Bee Lim, Stella Tsotsi, Chin-Ying Stephen Hsu, Sue Anne Toh, Swee Chye Quek, Victor Samuel Rajadurai, Walter Stunkel, Wayne Cutfield, Wee Meng Han, Wei Wei Pang, Yap-Seng Chong, Yin Bun Cheung, Yiong Huak Chan, and Yung Seng Lee.

## Author contributions

S.C., Z.H., J.K.Y.C., Y.S.C., and S.Y.C. proposed the original research question. J.Y.H. proposed additional hypotheses; linked, cleaned, and analyzed the data; wrote the manuscript; and synthesized final revisions. M.T.T., W.L.Y., I.M.A., K.M.G., N.K., L.Y.S., and J.G.E. contributed substantially to revisions, All authors approved of the manuscript and agree to be accountable for the author's own contributions and ensure all questions related to the accuracy and integrity of any part of the work are appropriated investigated, resolved, and resolution documented in the literature.

## Competing interests

Y.S.C., S.Y.C. and K.M.G. have received reimbursement for speaking at conferences sponsored by companies selling nutritional products, and are part of an academic consortium that has received research funding from Abbott Nutrition, Nestec, and Danone. The remaining authors declare no competing interests.
