## [Peer Review File · Nature Communications]

Reviewer #1 (Remarks to the Author):

Huang et al. examined the association between ART and shorter stature by multiple pathways including perinatal and early environment and DNA methylation. They observed that ART singletons (n=83) were shorter than non-ART singletons (n=1095) at around 6 years of age using data collected as part of GUSTO, a pregnancy cohort from Singapore. ART children also had lower skinfold thicknesses and blood pressure. These differences were not explained by parental anthropometry or other factors including epigenetics. While the manuscript is well written, some careful justification of the authors' analytic approaches is needed:

Major comments

- 1) Line 74-75, the authors should incorporate: Novakovic B et al. Nat Commun. 2019 Sep 2;10(1):3922. doi: 10.1038/s41467-019-11929-9. Assisted reproductive technologies are associated with limited epigenetic variation at birth that largely resolves by adulthood.
- 2) Please clarify the statement that begins (line 86) "We triangulate..." After re-reading it several times it is still unclear what are the strategies and what are the assumptions for each strategy and separately what is the differential treatment probability?
- 3) It's also not clear how the authors could expect meaningful (precise) estimates from analyses delineated from lines 90-98 to estimate true causal effects as in a pragmatic trial design while using observational data and mediation with a sample size of 83 ART children (7% of their data and actually even fewer according to Suppl Table 10 by 60 months, n~60, <75%). Hence, it is not surprising in some sense that all their follow-up analyses could not explain their findings of smaller size.
- 4) Their use of methylation data was also a little confusing. Prior to reading the methods and just from what was in the introduction, thought the authors were going to show differences by ART. But it seems according to lines 531-538 they searched the literature for CpGs associated with small size and came up with over 200 CpGs that fits this category. So rather than examining all 450K CpGs from their data, they limited their ART associations in supplemental Table 5 to only these CpGs, identifying NECAB3.
- 5) Some suggestions on figures.
 - a. Figure 1 was used to state that differences in height emerged 18-24 months – this seems to be driven by a beginning long right tail (maybe the graphs were just too crowded)? It is very difficult to say for sure why the authors picked that time point from these graphs. Perhaps it is not necessary to show these raw distributions when Figure 2 shows the associations. Although these seem to be starting at Birth (height-for-age z-score) so really they were just born shorter and persisted to be shorter (except for a random 6 month time point)? And it's a little weird at 18 months was the lowest response rate of ART (55 kids attended).
 - b. Can Supplemental Figure 2 be combined with supplemental Figure 3?
 - c. Supplemental Figure 5, perhaps move to primary material as this was an important part of the analysis to show that subfertility has similar patterns and that associations with ART may be due to confounding.
 - d. Supplemental Figure 6, not warranted/justified.
 - e. Supplemental Figure 8, more isn't always better, now it's just adding more and more comparisons of characteristics between groups with very small sample sizes of ART. As mentioned above, should drop mediation by postnatal factors as part of the analysis to better streamline the paper and not get lost in all these extraneous analyses that are not that meaningful even as supplemental information.
 - f. Supplemental Figure 9 seems to be one of the primary analyses that made this paper novel but seems buried?

Minor comments

- 1) It is lost upon me the link between line 31 and reference #21: Formenti F, Emmanuel Y, Constantin-Teodosiu D. 2010. Regulation of Human Metabolism by Hypoxia-Inducible Factor. PNAS 107 (28): 12722–7. The article is about hypoxia through muscle biopsies and nothing to do with fertility nor epidemiologic methods for a pragmatic trial.
- 2) Figure 4 – the color coding for ART and SC doesn't make sense for the Manhattan plot (part A) since those are p-values of the CpGs and have nothing to do w/ exposure status. But if it does have something to do with exposure status then y-axis is mislabeled.

Reviewer #2 (Remarks to the Author):

nice paper
important contribution to ART field

Reviewer #3 (Remarks to the Author):

The authors have investigated the effect of assisted reproductive technologies on long-term health of the offspring. They explored data from a prospective cohort of 83 assisted reproductive technologies vs. 1095 spontaneous singleton pregnancies. A number of pre-pregnancy and pregnancy factors including 187 epigenetic markers and obesity polygenic risk scores as well offspring outcomes at various ages up to 5 years of age were investigated. The study indicates that children conceived by assisted reproductive technology are shorter, have lower weight, skinfold thickness, and blood pressure. They observed no strong effect on metabolic biomarkers. Finally, they report that assisted reproductive technology was associated with reduced fetal cord tissue methylation at one site located within the NECAB3 (N-terminal EF-hand calcium-binding protein 3) gene.

Children conceived by medically assisted reproduction face an elevated risk of adverse birth outcomes. More knowledge is needed to establish the extent to which this excess risk is attributed to harmful effects of treatment or to pre-existing parental characteristics that confound the association. The present study is therefore welcomed. The data are very interesting although there are some questions to be raised.

1/. There are many other studies performed that are eg. prospective, population-based and/or much larger. It is not clear where lies the novelty of the findings? It seems the key findings are the effect on height and weight at age 5, but the study does not provide new knowledge for what the reason(s) might be.

2/. Large population-based studies seem to be needed in order to be able to draw firm conclusions. The present study investigated a cohort of 83 offsprings from assisted reproductive technologies and the study seems somewhat underpowered.

3/. Other studies on offspring of assisted reproductive technology have been reported to have higher rates of preterm birth, abnormal fetal size, and birth defects. Can the authors explain the differences in their findings?

4/. The authors did not investigate multiple pregnancies. From other prospective studies it seems that the possible negative effect on the child-health outcomes mainly is due to the higher incidence of multiple pregnancies and not because of assisted reproductive technologies.

5/. Sometimes it is hard to follow the statistical methods used, eg. the references in the text to results from Table and Supplementary Table 1 where data are given but no statistical comparisons are revealed.

6/. It is unclear how the polygenic risk score for obesity were generated. Which genes/common variants from children were used?

7/. The report they used cross-validated, ensemble machine learning algorithms. Please, explain in more detail exactly what is meant and how were these analyses performed?

8/. The methylation finding is interesting. Can the authors explain the rationale for choosing fetal cord? Did you consider using cord blood as well? Has the finding been validated eg. with methylation specific restriction endonucleases analysis, pyrosequencing, methylation specific high-resolution DNA melting, or quantitative methylation specific polymerase chain reaction? Also, have the authors looked for parent-of-origin effects of this region?

9/. Are the values given in Supplementary Table 2 corrected for gestational age and sex where appropriate? And is blood pressure adjusted for height? That is very relevant since lower length and lower blood pressure at age 5 years are some of the main findings.

10/. How are the betas compared in Supplementary Table 3?

11/. The varying number of cases in Supplementary Table 3 is somewhat unclear.

12/. The introduction and discussion are partly overlapping.

Reviewer #1 (Remarks to the Author):

Huang et al. examined the association between ART and shorter stature by multiple pathways including perinatal and early environment and DNA methylation. They observed that ART singletons (n=83) were shorter than non-ART singletons (n=1095) at around 6 years of age using data collected as part of GUSTO, a pregnancy cohort from Singapore. ART children also had lower skinfold thicknesses and blood pressure. These differences were not explained by parental anthropometry or other factors including epigenetics. While the manuscript is well written, some careful justification of the authors' analytic approaches is needed:

--- Thank you for the clear summary of some of our findings and your detailed review!

Major comments

1) Line 74-75, the authors should incorporate: Novakovic B et al. Nat Commun. 2019 Sep 2;10(1):3922. doi: 10.1038/s41467-019-11929-9. Assisted reproductive technologies are associated with limited epigenetic variation at birth that largely resolves by adulthood.

--- Novakovic, et al.'s finding that ART was associated with reduced cg03904042 methylation in both neonatal and adult blood directly supports our findings.

--- More broadly, their discussion of discordance in ART associations between fetal and adult blood epigenome supports our point that direct quantitative assessment of epigenomic mediation is essential in the context of non-persistent epigenomic signatures.

--- We are happy to incorporate discussion of their work.

Changes made as follows (changes in bold; line numbers refer to the original submission):

Introduction - Line 74: Recent EWAS show suggestive associations with ART, **but such associations may not persist longitudinally [Novakovic, et al]** and are likely confounded by infertility.

Discussion – Line 343-344: **An important recent study suggests that ART-fetal epigenome associations may not persist into adulthood [Novakovic, et al],** and no study has demonstrated whether **fetal changes, persistent or otherwise,** explain long term differences in child outcome.

Discussion – Line 358-359: **Most relevantly, a recent study by Novakovic, et al. found cg03904042 to be associated with ART status in both neonatal and adult blood, a site we replicated as part of our analyses ($\beta = -0.118$; p-value = 8.5×10^{-6} vs. $\beta = -0.097$, p-value = 2.1×10^{-9} , in our study) despite differences in platform (450k vs. EPIC) and tissue (fetal cord vs. neonatal blood).**

Discussion – Line 364: **However, as suggested by Novakovic, et al., the relevance of ART-related epigenomic changes to development is likely time-dependent. Consequently, the presence or absence of point-wise associations may not be sufficient to suggest functional relevance, and quantitative mediation analyses are necessary.**

2) Please clarify the statement that begins (line 86) “We triangulate...” After re-reading it several times it is still unclear what are the strategies and what are the assumptions for each strategy and separately what is the differential treatment probability?

--- Thank you for the opportunity to clarify what we believe to be a key strength of our paper.

--- We have now modified the statement to directly state the different design and estimation choices we used, and point the reader to the Methods section (see below) where the assumptions are more fully described.

--- We also expand the Discussion with a new stand-alone “Strengths of complementary design and estimation approaches” section to highlight this.

--- Importantly, in response to other comments, we have also added an additional set of complementary analyses using maternal mid-pregnancy peripheral blood (26 gestational week buffy coat) epigenome to test the assumption of residual confounding by unmeasured shared parental factors.

--- “Differential treatment probability” refers simply to the measured and unmeasured confounding factors which may influence both child outcome and their likelihood of receiving ART treatment, these include both underlying health/fertility factors as well as socioeconomic characteristics.

Changes made as follows:

Introduction – Line 86: We triangulate across several **complementary design (alternative comparison groups based on different maternal and paternal infertility risk factors, control for postnatal exposures, and negative control epigenetic mediators) and estimation (mixed effects, multiple imputation, inverse probability weighting, and doubly-robust estimation) strategies with different underlying assumptions (described in *Methods*) to estimate effects...**

Introduction – Line 98: **Because estimates of mediation are susceptible to residual confounding, particularly unmeasured maternal genetic and environmental factors that may influence both fetal epigenome and child growth, we conducted parallel simulations for mediation by maternal epigenome as negative controls for the presence of such confounding. Presence of substantial mediated effects through maternal methylation would provide evidence for residual confounding and against the hypothesis that observed effects are mediated through fetal genomic imprinting alone.**

Results – Line 251: **In simulation models testing for residual confounding, we found consistently estimated nulls for maternal NECAB3 (cg03904042) and HIF3A (cg27146050) methylation mediated effects (e.g. -0.0005 cm [-0.23, 0.23] and -0.03 cm [-0.20, 0.13], representing 0.01% and 1% of the total effect of ART on child height at 5.5 years; *Supplemental Figure 7*).**

Discussion - Line 369:

Strengths of complementary design and estimation approaches

To summarize, confidence in our findings were strengthened by triangulating across complementary design and estimation approaches: First, unbiased estimation of ART effects using observational data require appropriate selection of comparison groups. We demonstrated our

findings were robust to both subgroup restriction and standardization approaches (inverse probability weighting), with the former less subject to modeling choices and the latter taking better advantage of rich covariate data. Moreover, choosing different subgroup definitions reduced the possibility that measurement error for any particular set of measures (e.g. self-report vs. medical records) explained our findings.

Second, we investigated mediating mechanisms to strengthen evidence that observed ART-phenotype associations are due to proposed mechanisms (e.g. genomic imprinting) and not due to correlated exposures. Mediation analyses come at the cost of stronger assumptions of no unmeasured post-exposure confounding, which we addressed by assessing a negative control mediator -- maternal, mid-pregnancy DNA methylation status. If observed effects are mediated through fetal genomic imprinting alone, it follows that maternal epigenomic variation ought to have no independent effect on the phenotype. However, since maternal epigenome may be affected by ART, direct adjustment for maternal CpGs may result in biased estimates (collider stratification bias). We found no evidence of mediation through corresponding maternal loci which helps to reduce the chance of either unmeasured common confounders (of both maternal and fetal epigenome) or direct effects of maternal epigenome.

Third, we found our results were robust to complementary estimation techniques such as mixed effects, which assume random missingness, and multiple imputation, which depend on proper modeling of the missingness mechanism. Overall, we demonstrated that ART-phenotype associations are robust and that previously observed epigenomic findings may be related to unmeasured differences in underlying maternal/parental health or otherwise noncausal.

3) It's also not clear how the authors could expect meaningful (precise) estimates from analyses delineated from lines 90-98 to estimate true causal effects as in a pragmatic trial design while using observational data and mediation with a sample size of 83 ART children (7% of their data and actually even fewer according to Suppl Table 10 by 60 months, $n \approx 60$, $< 75\%$). Hence, it is not surprising in some sense that all their follow-up analyses could not explain their findings of smaller size.

--- In our target trial analyses, we found robust and consistent estimates of ART effects on reduced height and weight. However, it is generally true that mediation analyses are of relatively lower power than main effects analyses and we agree that power may be reduced to detect some differences, so we have clarified this point in the Discussion (see below).

--- Nonetheless, we first point out that we do find several CpG-mediated effects significant at the 0.05 level (e.g. $\sim 16\%$ of ART-associated weight-for-age reduction at 36 months mediated by cg03904042; -0.05 SD [95% CI: -0.01 , -0.004]) and that this precision was enabled by both multiple imputations for missing covariate data as well as efficiency gains from machine learning-based estimation (Targeted Maximum Likelihood Estimation).

--- Moreover, all estimates were fairly precise: For example, the lower 95% confidence bound for the cg27146050-mediated effect on height Z-score (Figure 5) indicate the data are unlikely to be consistent with more than an 11% mediated effect, supporting our qualitative interpretation of minimal mediation. In our newly presented analyses of potential mediation by maternal epigenome, we find the same approach allows for estimation of a fairly precise null, e.g. a maternal cg03904042-mediated effect of 0.01% [4.6%, -4.6%], suggesting that we would have been able to detect fetal mediated effects at half the magnitude of what was observed ($\sim 5\%$), given similar distributional properties of cord tissue and maternal blood methylation.

--- Second, when evaluating evidence for mediation, we were also concerned with the overall magnitude of the potential effects, whether they varied over time, and the relative bias reduction when comparing to the subfertile arm. To those ends, our result demonstrates fairly definitively that a) the magnitude of the mediated effect was small and consistent over time and b) attenuated (e.g. $16\% \rightarrow 7\%$ mediated) in the subfertile substudy.

Changes made as follows:

Discussion – Line 387: Moreover, while we found the magnitude of mediation to be generally small and further attenuated in subcohort analyses ($< 10\%$), we cannot exclude the possibility that these differences may have been significant at conventional alpha values with larger sample sizes.

4) Their use of methylation data was also a little confusing. Prior to reading the methods and just from what was in the introduction, thought the authors were going to show differences by ART. But it seems according to lines 531-538 they searched the literature for CpGs associated with small size and came up with over 200 CpGs that fits this category. So rather than examining all 450K CpGs from their data, they limited their ART associations in supplemental Table 5 to only these CpGs, identifying NECAB3.

--- Our primary goal was not to conduct a full exploratory ART EWAS, but to investigate whether potential candidates found previously in the literature may causally mediate the phenotypes observed in this longitudinal cohort. We have clarified this in the Introduction (see below).

--- We previously noted that the two strongest CpG hits in the set, cg14921437 and cg03904042, reached epigenome-wide significance of $p < 3.6 \times 10^{-8}$ (Line 224).

--- For this revision, we performed a full EWAS of all sites passing QC ($N = 336,684$) and found only 3 additional sites (cg09350387 (FASTKD), $p = 2.2 \times 10^{-8}$; cg04278794 (OR2C1), $p = 5.2 \times 10^{-9}$; and cg22560193 (APC2), $p = 3.2 \times 10^{-11}$) passing EWAS p-value thresholds. However, only NECAB3 had multiple CpGs meeting significance thresholds. We add this information in Discussion and the corresponding Manhattan plot as Supplemental Figure 6 (see below).

--- We will refrain from speculating further at this time since it is beyond the primary scope of the paper. Nonetheless, these hits are certainly worthy of further exploration at a later date using the methods outlined here.

Changes made as follows:

Introduction – Line 98: Given the potential for many hits to be non-causal, we specifically focused on EWAS-identified candidate CpGs previously associated with ART, maternal infertility risk factors, or child anthropometry.

Results – Line 225: While the intention was to follow-up previously reported candidate CpGs, for completeness we conducted an agnostic epigenome-wide association study ($N = 336,684$ CpGs; Supplemental Figure 6) which confirmed the two genome-wide NECAB3 hits, and suggested three other sites passing epigenome-wide significance (cg09350387 (FASTKD), $p = 2.2 \times 10^{-8}$; cg04278794 (OR2C1), $p = 5.2 \times 10^{-9}$; and cg22560193 (APC2), $p = 3.2 \times 10^{-11}$). For the remaining analyses, we focus on the a priori specified genes.

Supplemental Figure 6 - EWAS of ART status. CpGs significant at epigenome-wide level ($p < 3.6 \times 10^{-8}$) labelled. Adjusted for maternal ethnicity, age, parity, pre-pregnancy BMI, and child sex. NECAB3-annotating CpGs shown in black.

5) Some suggestions on figures.

a. Figure 1 was used to state that differences in height emerged 18-24 months – this seems to be driven by a beginning long right tail (maybe the graphs were just too crowded)? It is very difficult to say for sure why the authors picked that time point from these graphs. Perhaps it is not necessary to show these raw distributions when Figure 2 shows the associations. Although these seem to be starting at Birth (height-for-age z-score) so really they were just born shorter and persisted to be shorter (except for a random 6 month time point)? And it's a little weird at 18 months was the lowest response rate of ART (55 kids attended).

--- *We agree with the reviewer that differences occur over a range of time periods and vary between anthropometric measures and that is more clearly shown by Figure 2, which also account for covariates.*

--- *Thus, we have removed Figure 1 as suggested and reworded the relevant sentence (see below).*

--- *The lower participation at 18 months may have been a by-product of the visit schedule where certain visits may have been skipped due to the close spacing of visits (past visit at 15 months), if there was a delay in conducting the previous visit.*

Changes made as follows:

Results – Line 124-126: Lower weight and BMI and shorter height among ART children **tended to be more pronounced after 36 months, but patterns varied by anthropometric measure** (Figure 1).

b. Can Supplemental Figure 2 be combined with supplemental Figure 3?

--- *We have merged the two figures as requested*

c. Supplemental Figure 5, perhaps move to primary material as this was an important part of the analysis to show that subfertility has similar patterns and that associations with ART may be due to confounding.

--- *We have moved this figure to main results as Figure 3 and revised in-text citations as necessary.*

d. Supplemental Figure 6, not warranted/justified.

--- *Here we showed that reduced height amongst ART-conceived children was insensitive to alternative subfertile groups formed on the basis of different risk factors.*

--- *We can remove if necessary, but we feel this is valuable information demonstrating robustness against measurement error for several sets of independent measures (e.g. maternal medical records vs. paternal self-report).*

e. Supplemental Figure 8, more isn't always better, now it's just adding more and more comparisons of characteristics between groups with very small sample sizes of ART. As mentioned above, should drop mediation by postnatal factors as part of the analysis to better streamline the paper and not get lost in all these extraneous analyses that are not that meaningful even as supplemental information.

--- *We agree and have removed this figure.*

f. Supplemental Figure 9 seems to be one of the primary analyses that made this paper novel but seems buried?

--- *We have merged mediation of anthropometrics by fetal cord NECAB3 and HIF3A are shown together now in Figure 5.*

--- *The corresponding subcohort results for both are now also shown as Figure 6.*

Minor comments

1) It is lost upon me the link between line 31 and reference #21: Formenti F, Emmanuel Y, Constantin-Teodosiu D. 2010. Regulation of Human Metabolism by Hypoxia-Inducible Factor. PNAS 107 (28): 12722–7. The article is about hypoxia through muscle biopsies and nothing to do with fertility nor epidemiologic methods for a pragmatic trial.

--- *The citation refers to functional HIF defects resulting in deficits in cellular metabolism and aerobic capacity; this has potential ramifications for child physical fitness, musculoskeletal development, and therefore stature.*

--- *However, we agree that the specific experiments described therein may be a bit esoteric. We have replaced with a genetic epidemiologic study showing preliminary associations between maternal impaired glucose tolerance (an infertility correlate) and cord blood HIF3A methylation:*

Mansell T, Ponsonby AL, Januar V, et al. Early-life determinants of hypoxia-inducible factor 3A gene (HIF3A) methylation: a birth cohort study. Clin Epigenetics. 2019;11(1):96.

2) Figure 4 – the color coding for ART and SC doesn't make sense for the Manhattan plot (part A) since those are p-values of the CpGs and have nothing to do w/ exposure status. But if it does have something to do with exposure status then y-axis is mislabeled.

--- *We agree this is confusing and have now changed the color scheme for the Manhattan plot panel (A) to be different from the other two plots (B & C) and placed the exposure status legend directly on the plot.*

Reviewer #2 (Remarks to the Author):

nice paper

important contribution to ART field

--- Thank you for your supportive statement

Reviewer #3 (Remarks to the Author):

The authors have investigated the effect of assisted reproductive technologies on long-term health of the offspring. They explored data from a prospective cohort of 83 assisted reproductive technologies vs. 1095 spontaneous singleton pregnancies. A number of pre-pregnancy and pregnancy factors including 187 epigenetic markers and obesity polygenic risk scores as well offspring outcomes at various ages up to 5 years of age were investigated. The study indicates that children conceived by assisted reproductive technology are shorter, have lower weight, skinfold thickness, and blood pressure. They observed no strong effect on metabolic biomarkers. Finally, they report that assisted reproductive technology was associated with reduced fetal cord tissue methylation at one site located within the NECAB3 (N-terminal EF-hand calcium-binding protein 3) gene.

Children conceived by medically assisted reproduction face an elevated risk of adverse birth outcomes. More knowledge is needed to establish the extent to which this excess risk is attributed to harmful effects of treatment or to pre-existing parental characteristics that confound the association. The present study is therefore welcomed. The data are very interesting although there are some questions to be raised.

--- Thank you for the summary and the recognition of the outstanding challenges we aim to address directly in this paper: namely, disentangling the effects of assisted reproduction from underlying parental characteristics, including through better ways to interrogate molecular pathways. We hope that our revisions and added analyses further accentuate our strengths in this regard.

1/. There are many other studies performed that are eg. prospective, population-based and/or much larger. It is not clear where lies the novelty of the findings? It seems the key findings are the effect on height and weight at age 5, but the study does not provide new knowledge for what the reason(s) might be.

--- While there have been a few longitudinal studies, as evidenced by the Bay, et al. (2019) systematic review, results have been very inconsistent, investigate few time points within populations, and remain heavily confounded by parental characteristics. This is the first study that demonstrated a strong, consistent signal for both reduced height and weight over time and we provide some evidence for reduced blood pressure and thinness (via skinfolds). Further, as indicated by Novakovic et al. (2019), despite promising theoretical mechanisms such as defects in genomic imprinting, it is unclear whether previously observed epigenetic associations are empirical evidence for such mechanisms mediating child outcomes.

Our study improves on this literature by addressing both issues:

--- First, we believe our results add substantially to past work because we address parental confounding by several approaches ranging from sample restriction to direct adjustment for genetic, environmental, and paternal factors and accounting for selective attrition and missing data. We also demonstrate the trajectories are durable over time, rather than just 1 or 2 time points investigated by past work (e.g. Kai, et al. and Pontesilli, et al).

--- Second, we add clarity regarding the causal relevance of previously identified epigenetic associations and provide a careful way forward for querying epigenetic mediation hypotheses in

future molecular epidemiologic studies: First, we show evidence for potential mediation of child phenotype by previously described CpGs annotating to NECAB3 and HIF3A in fetal cord tissue. Next, we demonstrate a less biased estimate of mediation using our subcohort analysis (target trial approach) which greatly attenuated the estimated mediation.

--- Finally, in this revision, we introduce a negative control approach to investigate whether unmeasured confounding by parental characteristics may bias our estimates of epigenetic mediation. The new methods and finding are described below in reference to Question 8.

--- We feel having a reliable estimate of effects in non-randomizable settings, ruling out potential roles of previously describe candidate CpGs, and providing a roadmap for future explorations of epigenetic mechanisms are important and represents necessary progress in understanding long term effects of assisted reproduction, specifically, and other developmental hypotheses more broadly.

2/. Large population-based studies seem to be needed in order to be able to draw firm conclusions. The present study investigated a cohort of 83 offsprings from assisted reproductive technologies and the study seems somewhat underpowered.

--- *The reviewer raises two important points, first of replication of positive findings (presence of association) and second of lack of power in negative findings (absence of association). We address each in turn:*

Replication:

--- *First, our sample size is not unusual for assisted reproduction studies amongst deeply-phenotyped, longitudinal cohorts that enable investigation of mediating mechanisms. For example, our 83 ICSI-IVF subjects are more than well-known studies such as ALSPAC and Gen R (N ~60) with comparable measures. Larger longitudinal cohorts (e.g. ELFE) lack the-omic, rich covariate data, and number of follow-up time points available in this study.*

--- *Consequently, no other single study is currently able to provide a more detailed picture of this issue than ours. That our complementary analytic strategies demonstrating consistent associations over time and is supported by past findings with relatively strong designs (e.g. Kai, et al, Pontesilli, et al) lend strength to our claims. However, we look forward to working with other cohorts to collect new data and apply the full breadth of our approach to strengthen inference in the future.*

--- *Second, our findings with respect to cg03904042 are in fact directly replicated by Novakovic, et al. 2019 and part of our intention was to confirm or rule out epigenomic involvement in relevant mechanisms suggested by past EWAS discoveries (e.g. NECAB3 hits previously described and replicated). We discuss this in further detail below in response to Question 8.*

Power:

--- *We appreciate the concern that some of our models, particularly the mediation models, may be underpowered to detect differences. We have noted this limitation (Discussion – Line 387).*

--- *However, we point out that in our full cohort analyses, we do find nominally significant effects at the conventional $p < 0.05$ and moreover estimates had narrow confidence bands: For example, the lower 95% confidence bound for the cg27146050-mediated effect on height Z-score (Figure 5) indicate the data are unlikely to be consistent with more than an 11% mediated effect, supporting our qualitative interpretation of minimal mediation. In our newly presented analyses of potential mediation by maternal epigenome, we find the same approach allows for estimation of a null with narrow confidence bands, e.g. a maternal cg03904042-mediated effect of 0.01% [4.6%, -4.6%], suggesting that we would have been able to detect fetal mediated effects at half the magnitude of what was observed (~5%), given similar distributional properties of cord tissue and maternal blood methylation.*

3/. Other studies on offspring of assisted reproductive technology have been reported to have higher rates of preterm birth, abnormal fetal size, and birth defects. Can the authors explain the differences in their findings?

--- Incidence of preterm, low birth weight, and structural defects were indeed comparable between ART- and spontaneously conceived children in our cohort. This observation may be a joint consequence of the demographic characteristics of those who are able to access ART, our recruitment criteria (no Type I Diabetes Mellitus history, singletons only, willing to contribute biosamples), and prevailing ICSI-IVF protocols. We showed ART fetuses had similar estimated weight throughout pregnancy, so size at birth was not a chance finding. Moreover, maternal obesity and GDM, which are strong determinants of adverse pregnancy outcomes, were also well controlled in our pregnant population, e.g. with few exceeding recommended weight gain guidelines. We have added this consideration to our potential limitations.

Changes made as follows:

Discussion – Line 378: Moreover, on the basis of our study context and inclusion / exclusion criteria (e.g. no Type I Diabetes Mellitus history, willingness to contribute biosamples), our couples may be healthier overall than others previously studied. Despite having higher weight and blood pressures, for example, ART mothers had well-controlled gestational diabetes and few women exceeding gestational weight gain guidelines. While this obviously doesn't preclude future adverse outcomes in children, it was not observed with respect to immediate pregnancy outcomes.

4/. The authors did not investigate multiple pregnancies. From other prospective studies it seems that the possible negative effect on the child-health outcomes mainly is due to the higher incidence of multiple pregnancies and not because of assisted reproductive technologies.

--- The consideration of singletons only is a strength of our study as the different intrauterine milieu and post-natal feeding environment of multiples do not then contribute to our findings. It is also most relevant for the future given the trend towards single embryo transfers and fewer multiple pregnancies. Most other recent studies also take the approach of investigating singletons only or disaggregating. Considering that twins are often smaller in stature in utero and infancy, we would speculate that such effects may be even stronger, but this should certainly be investigated further such as through birth cohort consortia.

5/. Sometimes it is hard to follow the statistical methods used, eg. the references in the text to results from Table and Supplementary Table 1 where data are given but no statistical comparisons are revealed.

--- *We present the relevant measures of central tendency and variability in these tables merely to describe the population. No statistical tests are performed, which follows best practices for non-experimental studies as laid out by STROBE guidelines (Vandenbroucke, et al. 2007). This is because there is no a priori belief that variables will be jointly balanced between exposure groups in observational studies (and chance imbalances will occur in corresponding trials). Following best practices, strong a priori confounders will be included in adjustment models regardless of the p-value corresponding to this null. Instead, only the magnitude and variability of estimates from final adjusted models should be used to infer beyond the observed population.*

Vandenbroucke JP, von Elm E, Altman DG, et al. Strengthening the Reporting of Observational Studies in Epidemiology (STROBE): explanation and elaboration. PLoS Med. 2007;4(10):e297.

6/. It is unclear how the polygenic risk score for obesity were generated. Which genes/common variants from children were used?

--- *The baseline set of variants for the PRS were taken from the 97 genome-wide significant loci identified within the GIANT consortium GWAS on adult obesity (Locke, et al. 2015). For each of the three reported ethnicities, several PRS were generated on the basis of subsets of these 97 SNPs. Namely, they were reduced by proximity (250-kb & 5-Mb), putative Linkage Disequilibrium ($R^2 > 0.5$ & 0.2), and p-values. A range of p-value thresholds were used to form SNPs subsets. For each, a PRS was constructed for each child by weighting the number of risk alleles of each SNP by the GIANT-reported regression coefficient and standardizing within-ethnicity scores to mean 0 and variance 1. To choose a final PRS amongst those constructed, the p-value threshold (i.e. SNP subset) producing PRSs with the strongest correlation with offspring birth BMI was selected for each ethnicity separately. Thus, different SNP sets (and p-value thresholds were chosen for Chinese, Malay, and Indian offspring. This has been clarified in the Methods section (Line 552).*

Changes made as follows:

Methods – Line 552: For each ethnicity, the p-value threshold producing a PRS with the strongest correlation to child birthweight and BMI were retained (0.5, 0.1, and 10^{-4} , for Chinese, Malay, and Indian, respectively).

Locke AE, Kahali B, Berndt SI, et al. Genetic studies of body mass index yield new insights for obesity biology. Nature. 2015;518(7538):197-206.

7/. The report they used cross-validated, ensemble machine learning algorithms. Please, explain in more detail exactly what is meant and how were these analyses performed?

--- We have expanded the relevant Methods section (line 589-607) by drawing out into a separate section and giving more detail on how doubly-robust, cross-validated, ensemble machine learning was implemented within the targeted maximum likelihood estimation (TMLE) framework; the rationale; relevant assumptions; and additional introductory references that we hope will be helpful.

The section now reads as below:

Doubly-robust ART effect estimation with cross-validated, machine learning

Fourth, because both outcome modelling (approach one; covariate adjustment) and treatment modelling (approach three; propensity score weighting) approaches may be subject to misspecification of either the covariate set or functional forms, we implemented targeted maximum likelihood estimation (TMLE), which is doubly robust in that the estimator is unbiased if either model is correct and bias decreases as a function of the bias of both outcome and treatment models, otherwise (Rotnitzky, et al. 2019). Importantly, the use of such doubly-robust estimators enable the implementation of semi- and non-parametric machine learning estimators (Benkeser, et al. 2017), whereas their usage in standard (“singly robust”) outcome regression and propensity score settings may in fact lead to greater bias. (Naimi, et al. 2020). For this study, Collaborative-TMLE (C-TMLE; *ctmle* package in R) was implemented to prevent overfitting in the setting of sparse treatment models (Ju, et al. 2019).

For each model, an initial outcome regression was fit using the same ART exposure and covariates as approach one (multivariable regression), but with outcome Y scaled to [0,1]. An intercept-free logistic regression was fit repeatedly on Y using the predicted value from step 1 as an offset and an iteratively-updated propensity score as the only covariate. The propensity score was iterated by sequentially adding covariates to minimize the error in predicting Y. Once a minimum is found, the average treatment effect (ATE) was computed by taking the inverse-logit (expit) of the sole parameter of this final (influence function) regression and back-converting to the original scale. Standard errors (SE) were calculated from the variance of this influence function and Wald-based 95% confidence intervals constructed by $ATE \pm 1.96 * SE$.

Notably, this estimator can be used with standard regression approaches for the initial outcome and propensity score estimations without employing non-parametric algorithms (machine learning). However, as mentioned above, a key strength of this estimator is the ability to incorporate machine learning (coupled with cross-validation to prevent overfitting) to reduce residual confounding in both the outcome or treatment models. To this end, all outcome and propensity score predictions were estimated via SuperLearner (a clear introduction is given by Naimi & Balzer 2018): For each model, the data were split into 5 equal parts with 4 used for fitting a predictive model (training) and 1 held out for estimating fitted values (testing). A library of algorithms was fit to the training data including: GLM, GLM with interactions, Bayes GLM, observed means, neural net with a single hidden layer, and XG Boost (boosted trees). For each algorithm, fitted values are computed 5 times, each time taking a different combination of 4 training parts and 5th test set, and then averaged. This results in a set of cross-validated predicted values for each algorithm. To compute the contribution of each algorithm to the final model, a model (non-negative least squares) is fit regressing the observed outcome by the predictions of each algorithm. The coefficients for this fit then become weights for the prediction of each algorithm with better performing algorithms effectively contributing more to the prediction. Finally, the set of algorithms are fit to the original dataset and their fitted values are weighted by these estimated coefficients to form the final best prediction.

8/. The methylation finding is interesting. Can the authors explain the rationale for choosing fetal cord? Did you consider using cord blood as well? Has the finding been validated eg. with methylation specific restriction endonucleases analysis, pyrosequencing, methylation specific high-resolution DNA melting, or quantitative methylation specific polymerase chain reaction? Also, have the authors looked for parent-of-origin effects of this region?

--- *Our choice of fetal cord tissue was motivated by the goal of evaluating the tenability of genomic imprinting hypotheses as an explanatory mechanism for persistent effects of ART on child stature. We have previously shown in a substudy that while fetal cord and cord blood are generally concordant in estimated effects sizes and both meaningful surrogates of relevant mesodermal origin tissues, e.g. musculoskeletal, cardiac, circulatory, etc. fetal cord methylation showed greater inter-individual variability and was less susceptible to cis-SNPs (Lin, et al. 2017).*

--- *While we have not conducted any direct sequencing to confirm array findings, we note that our strongest hit cg03904042 is directly replicated in cord blood in the recent ART EWAS by Novakovic, et al. (2019). Also, Partida, et al.'s (2018) recent investigation of PoE effects suggests that this and other NECAB3-annotating CpGs may be imprinted as part of a block including the known imprinted MCTS2P pseudogene / HM13 gene (20q11.21) differentially methylated region.*

--- *Consequently, we were less concerned about whether such signals existed in fetal-derived tissues as the bigger question as to whether such associations were demonstrably relevant to the imprinting hypothesis to warrant follow-up confirmatory experiments. Specifically, we were very interested in what other information we could glean from interrogating non-fetal tissues. To that end, we investigated associations with corresponding sites in maternal mid-/late- pregnancy peripheral blood (buffy coat drawn at 26-gestational week visit) as negative controls to triangulate against alternative explanations (Lawlor, et al. 2016), and those analyses are now described in this manuscript.*

--- *Under the working hypotheses that fetal genomic imprinting is the explanatory mechanism: the maternal epigenome should not be related to offspring phenotype independently of intrauterine environment and fetal epigenome. Presence of an effect of corresponding maternal CpGs therefore implies either common unmeasured confounding (i.e. by parental morbidity or genetic contributions) or a direct effect of maternal epigenome, both of which would weigh against an imprinting hypothesis. Interestingly, we found very precisely null mediated effects by maternal epigenome when pregnancy complications are taken into account in our mediation simulation framework, suggesting our results are likely unconfounded.*

Lawlor DA, Tilling K, Davey Smith G. Triangulation in aetiological epidemiology. Int J Epidemiol. 2016;45(6):1866-1886. doi:10.1093/ije/dyw314

Lin X, Teh AL, Chen L, et al. Choice of surrogate tissue influences neonatal EWAS findings. BMC Med. 2017;15(1):211. Published 2017 Dec 5. doi:10.1186/s12916-017-0970-x

Novakovic B, Lewis S, Halliday J, et al. Assisted reproductive technologies are associated with limited epigenetic variation at birth that largely resolves by adulthood. Nat Commun. 2019;10(1):3922.

Partida GC, Laurin C, Ring SM, et al. Genome-wide survey of parent-of-origin effects on DNA methylation identifies candidate imprinted loci in humans. Hum Mol Genet. 2018 Aug 15; 27(16): 2927–2939.

Additions have been made as follows:

Introduction – Line 98: *Because estimates of mediation are susceptible to residual confounding, particularly unmeasured maternal genetic and environmental factors that may influence both fetal epigenome and child growth, we conducted parallel simulations for mediation by maternal epigenome as negative controls for the presence of such confounding. Presence of substantial mediated effects through maternal methylation would provide evidence for residual confounding and against the hypothesis that observed effects are mediated through fetal genomic imprinting alone.*

Methods – Line 688:

Negative control analyses

Under a putative fetal genomic imprinting hypothesis, maternal epigenome ought to have no direct effects on offspring phenotype when pregnancy comorbidities are accounted for (see causal diagram). Thus, if maternal CpGs mediated effect is detected independent of maternal complications this indicates either: residual confounding by unmeasured parental characteristics (estimates for fetal epigenetic mediation are biased) or there is a direct effect of maternal epigenome (fetal imprinting hypothesis is inadequate). To evaluate this, we recompute the mechanistic simulations using corresponding maternal mid-pregnancy methylation residuals in place of the fetal cord measures. Moreover, to test the sensitivity to temporal and causal ordering of the indicated diagram, we also compute the mediated effect after reordering our system of equations as follows (where M_CPG indicate the maternal methylation value):

$$E[M_CPG] = \beta_1(ART) + \Gamma(\text{confounders}) + \alpha_1$$

$$E[FG] = \beta_1(ART) + \beta_2(M_CPG) + \Gamma(\text{confounders}) + \alpha_2$$

$$E[SBP] = \beta_1(ART) + \beta_2(M_CPG) + \beta_3(FG) + \Gamma(\text{confounders}) + \alpha_3$$

$$E[GA] = \beta_1(ART) + \beta_2(M_CPG) + \beta_3(FG) + \beta_4(SBP) + \Gamma(\text{confounders}) + \alpha_4$$

$$E[Y] = \beta_1(ART) + \beta_2(M_CPG) + \beta_3(FG) + \beta_4(SBP) + \beta_5(GA) + \Gamma(\text{confounders}) + \alpha_5$$

The computed natural indirect effect (NIE) therefore should correspond to zero if both of our assumptions about unmeasured confounding and no-direct effect of maternal CpGs is correct and non-zero if either are violated. We found fairly precise estimates of null mediation by maternal CpGs which was invariant to causal ordering, strengthening our confidence that our estimates of fetal cord tissue mediation are unbiased, if small and potentially underpowered.

Negative Control Mediation by Maternal DNA methylation (DNAm). This diagram illustrates the causal relationships between ART, maternal and fetal epigenome, and offspring phenotype implied by a fetal genomic imprinting hypotheses. Notably, if maternal underlying health, ART status, and pregnancy complications are appropriately controlled for (*indicated by red lines*), there should be no direct effect of Maternal DNAm on offspring phenotype, and accordingly the proportion mediated by Maternal CpGs should be zero (*as indicated by an absence of a solid blue path from ART through Maternal DNAm to Offspring Phenotype*). If a mediated effect of Maternal DNAm is non-zero this indicates either (a) the presence of unmeasured confounders (*dotted edges*) or (b) a direct effect of maternal DNAm (*dashed edge*).

Results – Line 251: *In simulation models testing for residual confounding, we found consistently estimated nulls for maternal NECAB3 (cg03904042) and HIF3A (cg27146050) methylation mediated effects (e.g. -0.0005 cm [-0.23, 0.23] and -0.03 cm [-0.20, 0.13], representing 0.01% and 1% of the total effect of ART on child height at 5.5 years; Supplemental Figure 7).*

*Supplemental Figure 7 – Negative control analyses of mediation by *cg03904042* (A) and *cg27146050* (B) methylation in maternal mid-pregnancy peripheral blood, target trial subcohort. Covariates and approach were identical to the fetal cord tissue methylation mediation simulations amongst the subfertile subcohort (target trial). Shaded regions represent mediation by top *NECAB3*-annotating hit (*cg03904042*) and top *HIF3A*-annotating hit (*cg27146050*) in panels (A) and (B), respectively.*

9/. Are the values given in Supplementary Table 2 corrected for gestational age and sex where appropriate? And is blood pressure adjusted for height? That is very relevant since lower length and lower blood pressure at age 5 years are some of the main findings.

--- Supplemental Table 2 give the raw distributions as observed; as with Table 1 and Supplemental Table 1, and following STROBE guidance, the intent is to describe the data and not provide inference (which require additional adjustments as noted).

--- For all analyses to estimate differences, models are indeed adjusted for both child sex and gestational age at delivery and confidence intervals (and indicators of nominal significance) given.

--- All blood pressure analyses are conducted in both raw units (mmHg) and standardized to exact age, sex, and height as per American Academy of Pediatrics guidelines.

10/. How are the betas compared in Supplementary Table 3?

--- Supplemental Table 3 provides the crude and adjusted mean difference (units given in parenthesis) for each phenotype comparing ART versus spontaneously conceived (baseline). For example, ART children had ~1 kg lower mean fat mass at 5 years (beta = -0.9, [95% CI: -1.8, -0.04]) which was significant at a p-value < 0.05 threshold.

11/. The varying number of cases in Supplementary Table 3 is somewhat unclear.

--- Not all outcomes were measured in each child at each time point. For example, only 247 and 379 children underwent MRI for body composition at 5 and 6 years, respectively. For clinical biomarkers, observations may have been excluded for insufficient sample quantities per assay.

12/. The introduction and discussion are partly overlapping.

--- We have modified the introduction to focus more on the novelty of the proposed methodology. For example:

Introduction – Line 86: We triangulate across several complementary design (alternative comparison groups based on different maternal and paternal infertility risk factors, control for postnatal exposures, and maternal epigenome as negative controls) and estimation (mixed effects, multiple imputation, inverse probability weighting, and doubly-robust estimation) strategies with different underlying assumptions (described in Methods).

Introduction – Line 98: Because estimates of mediation are susceptible to residual confounding, particularly unmeasured maternal genetic and environmental factors that may influence both fetal epigenome and child growth, we conducted parallel simulations for mediation by maternal epigenome as negative control tests for the presence of such confounding. Presence of substantial mediated effects through maternal methylation would provide evidence for residual confounding and against the hypothesis that observed effects are mediated through fetal genomic imprinting alone.

Reviewer #1 (Remarks to the Author):

The authors have largely addressed my concerns, but some suggestions remain:

- The abstract mentions "simulations demonstrate ART status was strongly associated with lower NECAB3 CpG methylation..." but it is confusing to say it's non-causal when that was in reference to mediation not ART status correct? And then the next sentence is about the additional analyses requested by another reviewer but it just comes out of nowhere and turns to be able mediation. Maybe just simplify it to say something like while cord tissue NECAB3 methylation was associated with ART status, it does not mediate these differences in phenotype in childhood.
- Line 90-93 – was difficult to read such a long statement – please consider just speaking to design strategies and reword not saying "triangulate" which I don't think is the intended meaning here. It's much clearer in the text now in lines 440-446 – maybe can remove this long statement?
- Line 106 – consider stating the # of CpGs under consideration and at the least citing the references for picking these (187 CpGs). Or refer to Supplemental Table 5 for the citations. Although see below maybe just remove if focus on full EWAS findings.
- Line 248 – if this is the primary analysis (with the additional 94 HIF genes being justified) then perhaps it's not necessary for the previous paragraph regarding NECAB3. Even wonder if that is necessary given the authors did conduct the full EWAS now. It's just confusing as to why someone would essentially run 3 different sets of CpGs when they all say the same thing and is from the same dataset. Selective "looks" of the data should not give different results. It's the same conclusion without taking the readers through in such a roundabout fashion. And no need for Supplemental Table 5, unless they want to just show the previous literature on NECAB3. Maybe the additional information gleaned from running it the other 2 ways is lost upon me.
- Grammar
 - o Line 77 "may not persist" (not "persistent")
 - o Line 96 "probabilities" (not singular)
 - o Line 106 – candidate CpG sites or CpGs (need to be plural or add "a" before candidate if you are referring to only NECAB3)

Reviewer #3 (Remarks to the Author):

The authors have responded satisfactorily to my concerns. I have no further questions or comments.

Reviewer #5 (Remarks to the Author):

In this manuscript, the authors collected 83 children conceived by ART and 1095 spontaneously conceived controls. A second control set of 93 children was created after selection on parental medical records to match the ART parents. ART children had lower weight, height, BMI, skin thickness, blood pressure and fat mass, mostly observed after 36 months. The authors suggest these changes are independent of parental factors, although the effects diminish after comparing to the matched controls. Several of 281 tested candidate CpGs were significantly associated to ART status, and mediation analyses suggest these CpG signals might be independent. Overall, the observed changes in height, weight and BMI between ART and SC children appear robust and consistent across all analyses. The manuscript is well written and the results are clearly and completely described, although it can be hard to keep track of all the analyses performed.

1) The authors claim that parental factors do not impact the height, weight and BMI differences between ART and SC children. However, the effect estimates of all analyses seem to diminish when comparing to the parents selected as sub fertile trial-targets. Therefore, potential selection bias on the ART parents cannot be excluded without more evidence. This could be stated more clearly in the discussion.

2) It should be noted that many different tests have been performed. Although I agree that the observation of decreased size in ART children seems consistent, we should be wary of over testing and/or over interpretation of a relatively small sample size. The authors do not discuss multiple testing across their phenotypes, and formal replication of the main result is lacking, even if evidence from previous literature suggests these signals holds (studies which the authors agree are limited). By the end of the study, only 59 ART children remain, and the whole result could be driven by a random sampling unrelated to the ART-effect on development. It should be clearly stated that these findings are not conclusive until studied in a larger cohort.

3) The authors discussed that ART might impact child size, but does not seem to affect cardiometabolic outcomes. It is unclear what the authors suggest should be done with this information: should ART children be monitored or treated differently? How does this tie in to daily practice, and other considerations surrounding that aspect? The added value and novelty of this result is not immediately clear, and would permit relevance for a greater audience if this context/conclusion is more clearly included.

Minor points

1. Table 1: consider to also add characteristics of the selected controls to this table.
2. Table 2: please indicate also which test would pass multiple testing correction.

Best regards,
J. van Rooij, PhD

EDITOR'S COMMENTS

Please note that it is essential to address the comments raised by the reviewers, and in particular, the limitations linked to cohort size highlighted by reviewer #5 should be clearly acknowledged by modifying the text of the manuscript. Please also ensure that the suggestions from reviewer #1 are incorporated.

To respond to both comments from reviewer #5, we have added a final concluding paragraph providing context for interpretations given the limitation of the small, but typical, sample size and called for future work applying these methods in larger samples (e.g. consortia of suitable studies).

Reviewer #1 comments have also been incorporated. Notably, we clarify in-text that the EWAS was post-hoc (at reviewer request). We believe faithfully reporting the analyses as actually conducted is the best policy in this case. This provides appropriate context as to why, for example, we did not follow-up isolated, previously-unreported EWAS hits.

In addition, please clarify in the point-by-point response, and in the manuscript, how your study integrates in the general GUSTO framework (NCT01174875). For example, was the analysis presented pre-specified in the study protocol for the GUSTO study?

We have added the following to Methods (lines 550-553):

The GUSTO cohort study is registered under study ID: NCT01174875 (clinicaltrials.gov) which broadly covers investigations of parental and gestational influences on child health. The specific analyses conducted here were not pre-specified in the protocols.

We think that the advance provided by the manuscript resides mainly in the pragmatic trial approach using machine-learning estimators, and the title should emphasize this. I have suggested the following title: " Analysis of health related factors following assisted reproductive technologies using a pragmatic trial approach "

We have adopted the suggested title with slight modifications.

REVIEWERS' COMMENTS

Reviewer #1 (Remarks to the Author):

The authors have largely addressed my concerns, but some suggestions remain:

- The abstract mentions “simulations demonstrate ART status was strongly associated with lower NECAB3 CpG methylation...” but it is confusing to say it’s non-causal when that was in reference to mediation not ART status correct? And then the next sentence is about the additional analyses requested by another reviewer but it just comes out of nowhere and turns to be able mediation. Maybe just simplify it to say something like while cord tissue NECAB3 methylation was associated with ART status, it does not mediate these differences in phenotype in childhood.

We agree the edits were confusing. We have rephrased as suggested to:

“Our simulations demonstrate ART status is strongly associated with lower *NECAB3* DNA methylation, with negative control analyses suggesting these estimates are unbiased. However, methylation changes do not appear to mediate observed differences in child phenotype.”

- Line 90-93 – was difficult to read such a long statement – please consider just speaking to design strategies and reword not saying “triangulate” which I don’t think is the intended meaning here. It’s much clearer in the text now in lines 440-446 – maybe can remove this long statement?

We have simplified the sentence to say:

“In this study, we estimate effects of ART-assisted conception on child anthropometry and cardiometabolic outcomes using several complementary design and statistical modelling strategies with different underlying assumptions (described in *Methods*), taking into account differential treatment probabilities, residual confounding, and loss-to-follow-up.”

- Line 106 – consider stating the # of CpGs under consideration and at the least citing the references for picking these (187 CpGs). Or refer to Supplemental Table 5 for the citations. Although see below maybe just remove if focus on full EWAS findings.

We now provide the number in text (N = 281) and reference the Supplemental Table (now labelled Supplemental Data) which contain the full citations.

- Line 248 – if this is the primary analysis (with the additional 94 HIF genes being justified) then perhaps it’s not necessary for the previous paragraph regarding NECAB3. Even wonder if that is necessary given the authors did conduct the full EWAS now. It’s just confusing as to why someone would essentially run 3 different sets of CpGs when they all say the same thing and is from the same dataset. Selective “looks” of the data should not give different results. It’s the same conclusion without taking the readers through in such a roundabout fashion. And no need for Supplemental Table 5, unless they want to just show the previous literature on NECAB3. Maybe the additional information gleaned from running it the other 2 ways is lost upon me.

The reviewer is correct that the choice of subsets would not change the frequentist p-values as reported and we recognize that the description of iterative findings is complex.

For complete transparency, though, we want to present these analyses exactly how they were planned and conducted, which was driven by these *a priori* higher probability sites. This also allows us to describe and visualize those sites chosen from prior literature regardless of

magnitude or significance thresholds.

Moreover, since the EWAS was conducted post-hoc at reviewer's request and not an initial objective of our study, we did not follow up the other isolated CpG hits (cg09350387, cg04278794, and cg22560193) which have not previously been reported. This decision would be harder to justify if we had presented the EWAS as our primary/only analyses.

To clarify this, we rephrase the line to specifically say:

"While the original objective was to only follow-up previously reported candidate CpGs, at reviewer's request we additionally conducted an agnostic EWAS..."

- Grammar

o Line 77 "may not persist" (not "persistent")

Corrected as specified

o Line 96 "probabilities" (not singular)

Corrected as specified

o Line 106 – candidate CpG sites or CpGs (need to be plural or add "a" before candidate if you are referring to only NECAB3)

Corrected to "candidate CpGs"

Reviewer #3 (Remarks to the Author):

The authors have responded satisfactorily to my concerns. I have no further questions or comments.

Reviewer #5 (Remarks to the Author):

In this manuscript, the authors collected 83 children conceived by ART and 1095 spontaneously conceived controls. A second control set of 93 children was created after selection on parental medical records to match the ART parents. ART children had lower weight, height, BMI, skin thickness, blood pressure and fat mass, mostly observed after 36 months. The authors suggest these changes are independent of parental factors, although the effects diminish after comparing to the matched controls. Several of 281 tested candidate CpGs were significantly associated to ART status, and mediation analyses suggest these CpG signals might be independent. Overall, the observed changes in height, weight and BMI between ART and SC children appear robust and consistent across all analyses. The manuscript is well written and the results are clearly and completely described, although it can be hard to keep track of all the analyses performed.

1) The authors claim that parental factors do not impact the height, weight and BMI differences between ART and SC children. However, the effect estimates of all analyses seem to diminish when

comparing to the parents selected as sub fertile trial-targets. Therefore, potential selection bias on the ART parents cannot be excluded without more evidence. This could be stated more clearly in the discussion.

Despite the extensive design strategies to specifically minimize selection bias with respect to past observational studies, we agree that it remains possible as suggested.

We further emphasize the limitation by expanding on the following section (**additions underlined and bolded**):

“While we attempted to mitigate selection bias through inverse probability weighted models and alternate comparison groups, the possibility remains that the observed cohorts may not represent those enrolled in a pre-conception study which will need to be addressed in corresponding cohorts. Notably, while we included information on paternal age, anthropometrics, diabetes, and high blood pressure history and showed that differences persisted relative to a comparison arm of children of fathers with higher risk factors, we cannot rule out the possibility that specific unmeasured male infertility factors may underlie observed differences. **Additionally, measurement error for included covariates or other underlying maternal or familial characteristics not included in our modelling may contribute to residual confounding. Overall, we cannot rule out that better accounting for additional factors may further reduce observed effects.**”

2) It should be noted that many different tests have been performed. Although I agree that the observation of decreased size in ART children seems consistent, we should be wary of over testing and/or over interpretation of a relatively small sample size. The authors do not discuss multiple testing across their phenotypes, and formal replication of the main result is lacking, even if evidence from previous literature suggests these signals holds (studies which the authors agree are limited). By the end of the study, only 59 ART children remain, and the whole result could be driven by a random sampling unrelated to the ART-effect on development. It should be clearly stated that these findings are not conclusive until studied in a larger cohort.

We agree that the findings should not be over-interpreted and should absolutely be studied via consortia of cohorts (most international longitudinal cohorts with comparable measure have similarly sized IVF-ICSI populations as our study). We now add this contextualizing statement to a stand-alone *Conclusion* paragraph (lines 515-523):

“That said, while our results are internally consistent, they should not be over-interpreted and questions remain. Our study was enabled by the relatively high prevalence of ICSI-IVF treatment which resulted in case numbers comparable to much larger birth cohorts. Nonetheless, as a result of loss-to-follow-up and intensive measures, our effective sample sizes were modest in absolute terms and many exploratory analyses were conducted. Instead, we see these findings as a roadmap for strengthening future analyses, for example in consortia of parent-offspring cohorts with similar genotypic and phenotypic data. Application of these approaches to larger sample sizes, varied population and treatment protocols, and longer follow-up will be necessary for definitive conclusions.”

3) The authors discussed that ART might impact child size, but does not seem to affect cardiometabolic outcomes. It is unclear what the authors suggest should be done with this information: should ART children be monitored or treated differently? How does this tie in to daily practice, and other considerations surrounding that aspect? The added value and novelty of this result is not immediately clear, and would permit relevance for a greater audience if this context/conclusion is more clearly included.

We clarify the practical implications of findings, in light of the caution against over-interpretation, in the new *Conclusion* paragraph (lines 507-515):

“Conflicting current evidence on long-term cardiometabolic harms in ART-conceived offspring may influence treatment decisions for those considering assisted reproduction and provoke worry amongst those who have conceived and delivered through these technologies. Thus, it is critical to produce the best evidence possible given available data and impracticality of withholding ART completely. Our study clarifies past observational findings by focusing on effects observable in an ideal ICSI-IVF trial while providing further context for the relevance of previously observed epigenetic changes. Our findings support ICSI-IVF producing no adverse early child cardiometabolic outcomes in offspring of sub-fertile couples despite effects on stature, which should provide some reassurances to providers and families. That said...” [see passage above]

Minor points

1. Table 1: consider to also add characteristics of the selected controls to this table.

We replaced the “overall” column (which was redundant, given its similarity to all spontaneous conceptions) with the putative subfertile control group as requested

2. Table 2: please indicate also which test would pass multiple testing correction.

Added as requested. Length and weight differences (standardized and natural units) would pass a strict Bonferroni correction of p-value = 0.05/28 comparisons = 0.0018